# Neuronal variability reflects probabilistic inference tuned to natural image statistics

Dylan Festa [1], Amir Aschner[2], Aida Davila[2], Adam Kohn[1,2,3] & Ruben Coen-Cagli [1,2 ✉]

Neuronal activity in sensory cortex fluctuates over time and across repetitions of the same input. This variability is often considered detrimental to neural coding. The theory of neural sampling proposes instead that variability encodes the uncertainty of perceptual inferences. In primary visual cortex (V1), modulation of variability by sensory and non-sensory factors supports this view. However, it is unknown whether V1 variability reflects the statistical structure of visual inputs, as would be required for inferences correctly tuned to the statistics of the natural environment. Here we combine analysis of image statistics and recordings in macaque V1 to show that probabilistic inference tuned to natural image statistics explains the widely observed dependence between spike count variance and mean, and the modulation of V1 activity and variability by spatial context in images. Our results show that the properties of a basic aspect of cortical responses—their variability—can be explained by a probabilistic representation tuned to naturalistic inputs.

[1] Department of Systems and Computational Biology, Albert Einstein College of Medicine, Bronx, NY, USA. [2] Dominick Purpura Department of Neuroscience, Albert Einstein College of Medicine, Bronx, NY, USA. [3] Department of Ophthalmology and Visual Sciences, Albert Einstein College of Medicine, Bronx, NY, USA. ✉email: ruben.coen-cagli@einsteinmed.org

In sensory cortex, neuronal activity is typically variable, both in the absence of sensory input and for repeated presentations of a stimulus[1,2]. This variability is modulated by several sensory[3–11] and non-sensory[12–16] factors, suggesting it may play a functional role rather than simply reflecting noise. Understanding the functional role of variability is at the core of the inquiry of neural coding[17–22].

Parametric descriptive models can quantify how stimuli modulate neuronal variability[7,11,23,24], but they do not address why modulation of variability occurs and what functional role it might play. Here we develop and test a normative model, based on efficient coding[25–28] and probabilistic inference[10,29–33], to explain the properties of response variability in sensory cortex. In this approach, we hypothesize about functional and computational principles of cortical processing, to generate predictions about cortical activity. Specifically, we propose that probabilistic inference tuned to the statistics of natural images can explain the properties of response variability in visual cortex.

Although normative models have typically been used to explain trial-averaged responses, they can also be used to explain response variability[20,34–36]. In particular, some aspects of variability in primary visual cortex (V1) can be explained by the theory of neural sampling. This theory builds on the broader idea that the brain approximates operations of probabilistic inference[37,38], and hypothesizes that instantaneous neuronal activity represents samples from a probability distribution[20,34,39]. According to this view, variability of neuronal activity reflects uncertainty about the visual input (i.e., the width of the inferred probability distribution). As a result, variability is reduced by stimulus onset[4] and stimulus contrast[3,40], because of a reduction in uncertainty[10].

Here we hypothesize that modulation of uncertainty by visual input should reflect inferences tuned to the statistics of natural images, and thus the properties of response variability should reflect the statistical structure of images. To test this prediction, we consider a successful modeling framework, the Gaussian scale mixture (GSM[41,42]). This model assumes that images are composed by local features (e.g., oriented edges; Fig. 1a) and global features (e.g., image contrast), and that V1 neurons aim to represent the local features while discarding the global features[10,27,33,43,44]. GSMs can explain the modulation of trial-averaged V1 responses by stimuli in the surround of the receptive field (RF[45–50]). However, it is unclear whether this framework can also explain the surround modulation of variability[51,52] and whether this modulation reflects the statistical properties of natural inputs.

Here we combine modeling and electrophysiology in macaques to test our hypothesis that V1 variability is tuned to natural image statistics. First, we show analytically that the dependence between spike count variance and mean observed empirically[2,7,17] emerges in the GSM from the multiplicative interactions between local and global image features. Second, we show that stimuli in the RF surround modulate these interactions, and thus also response variability. Finally, we test predictions about surround modulation of firing rate and variability with recordings in V1 of awake and anesthetized macaques viewing natural images and gratings.

Our results show that visual context modulates neuronal response strength and variability independently, suggesting these modulations reflect probabilistic inference about local visual features. Our work thus provides evidence that the tuning of cortical variability can be explained assuming the brain performs operations of probabilistic inference of natural image statistics.

## Results

### The dependence between spike count variance and mean reflects multiplicative interactions between latent variables. To study the relation between natural image statistics and V1 cortical

variability, we considered the GSM because it captures the most prominent aspects of low-level image statistics, namely the sparseness of V1-like, oriented visual features and their nonlinear statistical dependence[27,41]. We assumed that the instantaneous firing of V1 neurons (Methods Eq. 4) represents samples from the inferred probability distribution (termed posterior distribution[10,31]) of oriented visual features encoded by the neurons. The inference of the posterior distribution requires inverting the so-called generative model of stimuli: that is, how features—small patches with different orientations and positions—are combined to produce images (Fig. 1a). Given an input image, model neurons then encode the inferred probability distribution of the coefficients of those features in the image. This is illustrated schematically for a vertical feature in Fig. 1b—top. The posterior distribution (middle column) in this case was broad with a large mean, indicating that the vertical feature was strongly present in the input image, though its precise coefficient was uncertain. Conversely, the image in Fig. 1b—bottom contains little evidence for the vertical feature, leading to a narrow posterior centered near zero. In the sampling framework, neuronal responses represent samples from this posterior distribution (Fig. 1b, right column). Thus, the variance of the spike count distribution (i.e., the neuronal variability) reflects the variance or width of the posterior, corresponding to the uncertainty about the coefficient of the encoded feature.

We studied whether, in the GSM, response variance depends on response mean, as observed in V1[2,7,17]. The GSM assumes $x = v \, \mathbf{g}$ where the sensory input $x$ is the result of local features $\mathbf{g}$ (the variables encoded by the neurons) multiplied by a global modulator $v$ (e.g., image contrast). To gain intuition about the mean–variance relationship of the model, we first considered the simplest formulation of a GSM, where $x$ and $\mathbf{g}$ are 1-dimensional. Although the expression relating these quantities—$x = v \, g$—is deterministic, knowledge of $x$ is insufficient to determine $g$, due to the unknown $v$. Computing the probability distribution of $g$ by accounting for the possible values of $v$ is a fundamental operation of probabilistic inference, called marginalization[38,53]. Crucially, because of the multiplication, both the inferred value of $g$ and its uncertainty (i.e., the mean and standard deviation of the posterior over $g$) are divisively related to $v$ (Eqs. 2, 3). For instance, assume we observed $x = 10$ and we inferred that $v$ is likely to be between 1 and 2 (Fig. 1c, dark blue), then by marginalization we would infer that $g$ is with high probability between 5 and 10 (Fig. 1c, light blue). If instead $v$ was inferred to be in the interval 4–5 (Fig. 1c, dark brown), then $g$ could only take values between 2 and 2.5, thus shrinking both in mean and variance (Fig. 1c, light brown). This example illustrates why a neuron whose responses reflect samples from the inferred distribution of $g$ should display a dependence between mean and variance in its response statistics. Note that this dependency is not linear, nor do mean and variance strictly follow each other as they would in a Poisson process. In general, the relative scaling depends on model choices, such as the uncertainty on the priors and, for high dimensional inputs, the stimulus structure (as explained in the next section). Notice too that if the mixer term $v$ were additive instead of multiplicative, then changes in its inferred value would only change the inferred mean of $g$, not its variance, leading to different predictions (Supplementary Fig. S1).

To validate this intuition more rigorously, we considered GSM inference on real images. As in past normative models[27,33,43,44], we implemented a GSM with oriented filters[54] spatially arranged to define both the RF of the model neuron and its surround (Fig. 2a; details in Methods). The model was trained on a large ensemble ($N = 10,000$) of natural image patches extracted from the BSDS500 database[55] (https://github.com/BIDS/BSDS500).

Given an input image, the visual inputs $\mathbf{x}$ (a vector) were determined by the activations of those filters applied to the

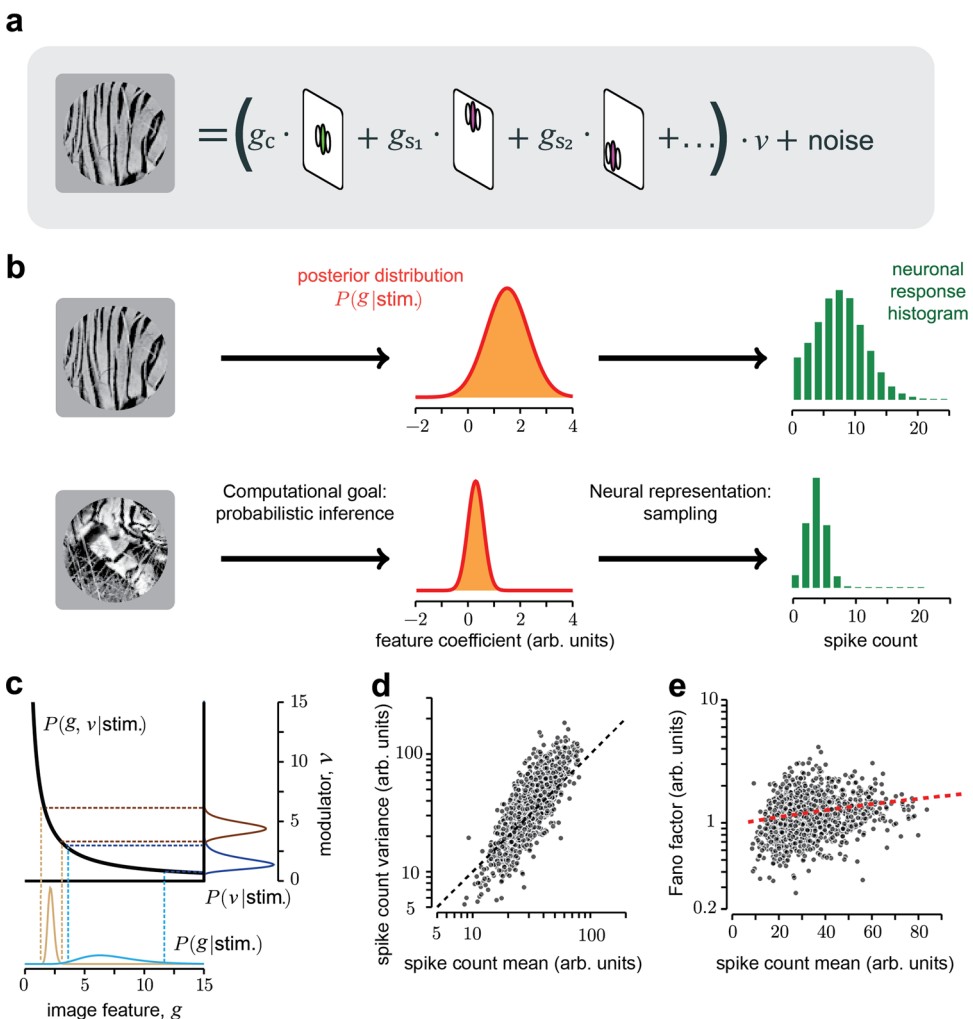

**Fig. 1 Sampling-based inference in the GSM model explains the dependence between spike count variance and mean. a** Representation of the generative process of the Gaussian scale mixture (GSM) model (Methods Eq. 1). The image (left) is described as the combination of local oriented features weighted by Gaussian coefficients, further multiplied by a global modulator and corrupted by additive Gaussian noise. **b** Encoding of sensory information according to the sampling hypothesis: the goal of a model neuron is to represent the posterior distribution (orange, middle) of the feature it encodes. The activity of the neuron corresponds to samples from that distribution, therefore the histogram of spike counts over time or repetitions (green, right) reflects the distribution. **c** Tuning of mean and variance in a 1-dimensional version of the GSM with no noise. For fixed input $x$, the visual feature $g$ and the modulator $\nu$ are bound to lie on the hyperbole $\nu = x/g$ (black line). Therefore, a larger estimate of $\nu$ implies reduced mean and variance of the posterior distribution of $g$ (blue versus brown curves). **d** Mean versus variance of a GSM model neuron in response to 1000 patches of natural images. Patches were selected randomly, with the requirement of sufficient signal strength inside the RF, i.e., above the median of the full distribution of $(x_{1+}^2 + x_{1-}^2)$ on natural scenes, where $x_{1+}$ and $x_{1-}$ are the odd and even phases of the center-vertical filter (see Methods). **e** The Fano factor (FF; ratio between mean and variance) as a function of the mean for the same GSM simulation reported in (**d**). Red dashed line represents the best linear fit. Pearson correlation coefficient 0.214, ($p < 10^{-4}$, two-sided $t$ test of the null hypothesis of zero correlation).

image. We denoted by **g** the corresponding local visual features. First, we verified that the multiplicative effect of the modulator allows the GSM to capture the statistics of natural images[41] better than an additive modulator (Supplementary Fig. S1). We found through analytical derivations and simulations that the variance of the inferred **g** grows with the mean, and both are divisively scaled by the estimate of the global modulator $\nu$, leading to a general reduction of uncertainty when the estimate of $\nu$ increases (Methods Eqs. 2, 3; see Supplementary Text for derivation). We then simulated model responses to a wide range of natural images (Fig. 1d), and characterized the mean–variance relation. The response variance of the model neuron scaled proportionally with its mean. Furthermore the ratio of variance to mean, termed Fano factor (FF), increased on average for stimuli that elicited stronger mean responses

(Fig. 1e), in qualitative agreement with the statistics of V1 neurons[7]. Importantly, training a GSM on different image sets, such as white noise, led to different parameter values but qualitatively similar predictions for neural responses (Supplementary Fig. S2), indicating that the mean–variance dependence arises from matching the generative model's structure to image statistics (i.e., multiplicative latent interactions) rather than fine-tuning its parameters.

These analyses confirm the intuition that the dependence between posterior variance and mean observed in the GSM emerges from the multiplicative interactions between the global modulator and the local variables. Because this partition between local and global variables in the GSM is known to capture well the statistics of natural images[27,41], our result establishes a precise link between image statistics and cortical variability.

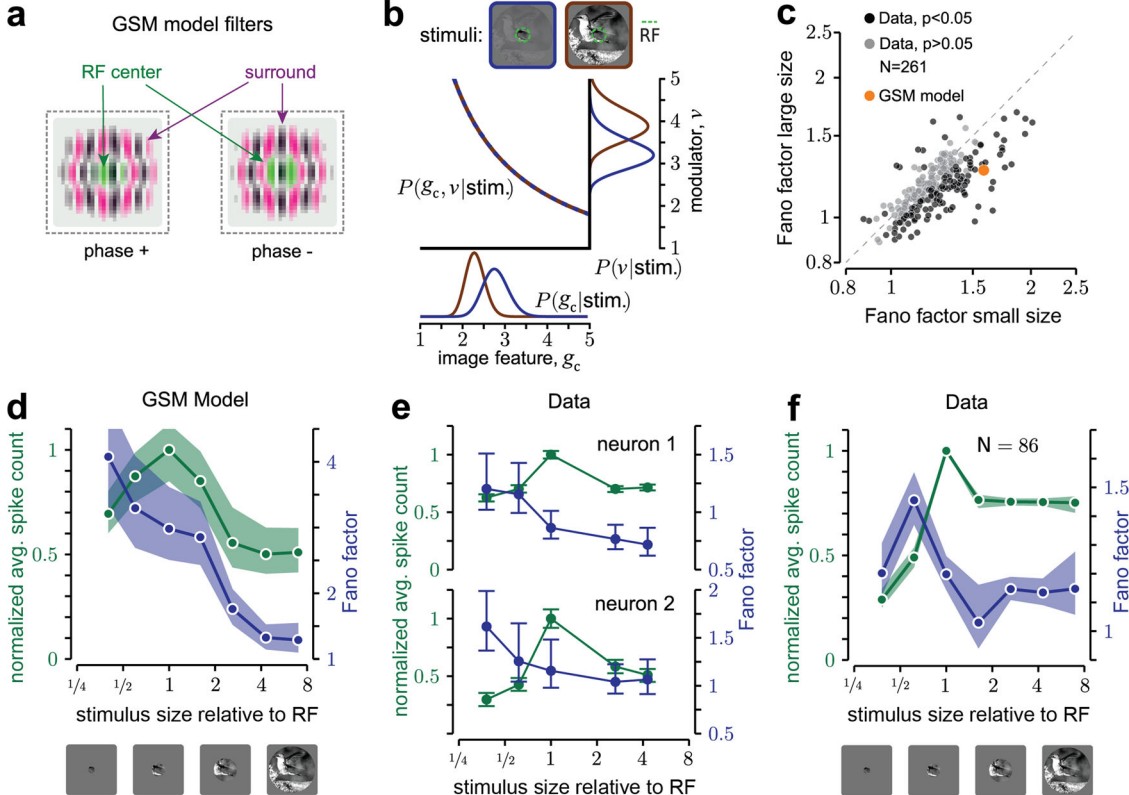

**Fig. 2 Surround stimulation reduces GSM uncertainty and V1 variability for natural images. a** In the GSM, the inputs to the model neuron are provided by the activity of quadrature pairs of oriented filters, corresponding to the spatial RF (green) and its surround (magenta). **b** Noise-free GSM model applied to an image without (blue) and with (brown) surround. The surround stimulus does not change the constraint between $g_c$ (the local feature associated with the RF center) and $\nu$, but it influences the estimate of the modulator and therefore also the estimate of $g_c$. **c** FF averaged across small (1°) and large (3.1° or 6.7°) natural image patches. Black and gray circles: average FF across images for each V1 neuron; black denotes a significant difference ($p < 0.05$) across the two conditions. Orange circle: average FF of the GSM response for the same set of images. For the conversion to spike counts (see Methods, Eq. 4) we used the scaling factor $c = 2$. The $p$ values were computed with a two-sided paired sample $t$ test, of the null hypothesis that the difference between the two conditions had mean equal to 0. **d–f** Tuning of the mean spike count (green) and FF (blue), for natural image patches of varying size. **d** GSM model, scaling factor $c = 15$. This constant was different than in (**c**), because the experiments of (**c**) used images with a broader range of orientation and frequency content than (**d**). **e** Data from one awake fixating macaque V1, for two example neurons and a single image presented at different sizes. Mean rate (green dots) and FF (blue dots) have been computed over 110 stimulus presentations. **f** Population average across neurons (86 in total) and image patches (10 in total). The error bars in (**d**, **e**, **f**) represent the 68% c.i. and are computed by bootstrapping.

**Surround stimulation reduces uncertainty and V1 variability.** The previous analysis shows that variability in the GSM is influenced by the inferred values of the global modulator. Therefore, the framework predicts that variability is sensitive to stimulus manipulations that affect the inferred global modulator. Specifically, stimuli that lead to a higher estimate of the modulator present less uncertainty over the hidden feature, and thus should reduce response variability. To test this prediction, we considered the modulation of V1 activity induced by spatial context—by stimuli in the surround of a neuron's RF—because spatial context can reduce stimulus uncertainty without modifying the stimulus drive inside the RF[56].

First, we verified for the GSM that surround stimuli (i.e., image regions that activate the surround filters) reduce uncertainty. The activity of the model neuron is associated with the oriented feature in the center. However, the surround input contributes to the estimate of the global modulator, and therefore influences the neuronal response. Specifically, our analytical results show that, for a fixed RF input, surround stimulation increased the estimated modulator and therefore had a suppressive influence both on the mean and variance of the neuronal response (Fig. 2b; Methods Eq. 3), validating our intuition that surround stimuli reduce

uncertainty because they result in a higher estimate of the global modulator.

Next, we tested whether surround stimulation reduces V1 variability, relative to RF stimulation alone, by analyzing previously published data on V1 surround modulation in anesthetized macaques[33]. In these experiments, natural image patches were presented at two different sizes, either masked to fit within the average RF (1°), or extending well beyond into the surround (3.1–6.7°). Among the neurons with a significant change in FF across conditions (127/261 neurons, $p < 0.05$), the vast majority (91.3%) had a lower FF for large images than small ones, consistent with model predictions. The average FF, across all neurons, was also lower for large images than small ones (1.15 versus 1.22, $p < 10^{-6}$, $N = 261$ neurons). We verified with a mean-matching analysis that this difference in FF could not be explained by differences in spike count mean (Supplementary Fig. S3). This result agrees qualitatively with the model (Fig. 2c, orange symbol), although surround suppression of FF was stronger in the model, possibly because surround modulation in the GSM is recruited by all images, whereas in V1 it is weak or absent for many images[33]. Consistent with this possibility, the strength of surround suppression of responsivity and of FF were positively correlated (Supplementary Fig. S4).

**Distinct effects of RF and surround stimulation on variability suppression**. Suppression of response variability by large stimuli might not be due solely to surround stimulation. Visual stimuli reduce the variability measured in spontaneous activity[4]. Therefore large images might reduce variability by providing stronger drive to the RF, in those cases where small images did not completely cover the RF. To test whether stimuli larger than the RF induced further reduction of the FF, beyond the reduction caused by the stronger RF drive, we considered responses to circular patches of natural images, with sizes ranging from much smaller to much larger than the typical RF.

We first studied the effects of stimulus size in the GSM. We found that the mean response peaked for images matched in size to the RF and decreased for larger stimuli, consistent with past work[44]. The FF, on the other hand, decreased monotonically with stimulus size, well after the stimulus filled the RF (Fig. 2d), because large stimuli lead to a larger estimate of the global modulator (Supplementary Fig. S5A). The difference between the behavior of the FF and the mean indicates that it should be possible to dissociate the effects of variability reduction from the modulation of spike count mean: stimuli smaller than the RF and larger than the RF can elicit similar average responses but with different variability.

We tested these predictions in V1 responses to natural images of different sizes in one awake fixating macaque. For the two example neurons of Fig. 2e, the mean spike count displayed the typical non-monotonic size dependence (green), whereas the FF decreased monotonically (blue). Similar effects were evident across all recorded neurons for stimuli ranging from approximately half the RF size up to several times larger (N = 86; Fig. 2f). The FF decreased by 18.7% as stimuli increased from ~½ RF size to RF size, and an additional 5.7% as stimuli increased from RF size to approximately twice that size (Table 1, left), which is the average extent of the suppressive surround in V1[46,48,49]. Furthermore, the FF decreased for stimuli larger than the RF compared to stimuli smaller than the RF, even when both stimuli evoked approximately the same number of spikes (Table 1, right). To be sure that our results were not affected by inaccurate estimates of RF size, due to variations in local contrast across natural images, we measured responses to static gratings in the same animal, and obtained similar results (Table 1, experiment 2; Supplementary Fig. S6A). New analyses of previously published data from anesthetized animals[33] also confirmed these results (Table 1, experiment 3; Supplementary Fig. S6B), ruling out the possibility that microsaccades in the awake animals might have introduced biases.

Note that the FF was lower on average for stimuli smaller than ½ RF size (Fig. 2f, leftmost point). This was true for the subset of neurons with large RF (N = 65/86), whereas the FF decreased strictly monotonically for neurons with smaller RFs (Supplementary Fig. S7). Both the large apparent RF size and the non-

monotonicity of the FF would be expected if stimuli were not perfectly centered on the RF (Supplementary Fig. S8). Furthermore, the FF decreased monotonically with stimulus size in the anesthetized dataset, for which stimulus centering could be controlled more tightly (Supplementary Fig. S7).

These analyses show that stimulation of the RF surround reduces response variability, beyond the known reduction from spontaneous to stimulus-driven activity[4].

**Surround suppression of variability is orientation selective**. Surround suppression of mean firing rate is known to be stronger for image patches with matched orientation inside and outside the RF, and weaker when the surround orientation is orthogonal to the center[45,47,49,57–59]. It is not known whether variability is similarly tuned. In our GSM model, surround tuning of mean responses (Fig. 3a, green) was obtained by using surround filters with the same orientation as the feature of interest inside the RF (details in Methods), as in past implementations[27,43].

Because the GSM predicts that surround suppression of both mean spike counts and variability is controlled by the inferred strength of the global modulator, we found that surround suppression of model response variability and of mean spike counts were similarly tuned (Fig. 3a). We verified that this corresponded to a smaller estimate of the global modulator for orthogonal surround stimuli (Supplementary Fig. S5B), which in turn resulted in weaker surround suppression of variability.

To test these model predictions, we measured V1 responses to compound static gratings in two awake, fixating macaques (N = 71 neurons). Consistent with past literature, the mean response was suppressed (relative to no surround) more when the surround and center orientations were matched (Fig. 3b; average suppression matched 0.844, orthogonal 0.885; average reduction 6.28%, p = 0.0043). In agreement with model predictions, the FF was smaller for the matched surround (Fig. 3c; average FF matched 0.973, orthogonal 1.02; average reduction 4.73%, p = 0.032), and this was true in the majority (N = 9/14) of neurons with a significant change (p < 0.05). However, although consistent with the GSM prediction, the magnitude of the effect was small (see also Discussion). One reason might be that, in our data, 26/71 neurons responded more strongly to matched than orthogonal surrounds (i.e., opposite to the surround tuning of our GSM implementation), which may be due both to imperfect stimulus centering and to the known heterogeneity in the orientation tuning of surround suppression of firing rate[49]. Consistent with this explanation, we verified that if we restricted our analysis to neurons that responded more weakly to matched than orthogonal surround (N = 45/71; average reduction 17.3%, p < 10^{-5}), the surround tuning of FF was also stronger (average reduction 7.37%, p = 0.013) than for the entire population (Supplementary Fig. S9).

---

**Table 1 Response variability decreases with stimulus size.**

| Experiment | FF decrease (½ RF)–(1 RF) | FF decrease (1 RF)–(2 RF) | p Value | Mean-matched FF decrease (size < RF)–(size > RF) | p Value |
|---|---|---|---|---|---|
| 1. Natural, awake (N = 86; Fig. 2f) | 18.7% | 5.7% | 0.0082 | 25.7% | <10^{-5} |
| 2. Gratings, awake (N = 19; Supplementary Fig. S4) | 31.7% | 9.0% | 0.05 | 47.7% | <10^{-3} |
| 3. Gratings, anesthetized (N = 229; Supplementary Fig. S4) | 14.2% | 7.0% | <10^{-3} | 22.6% | <10^{-5} |

*Rows*: separate experiments, with number of neurons selected in each experiment (inclusion criteria in Methods). *Columns*: Column 1, experiments. Columns 2–4, changes in FF with stimulus size. Columns 5 and 6, mean-matched (see Methods) change in FF with stimulus size. In all cases, a positive change denotes a reduction in FF for larger stimuli. Column 2: change in FF (Methods, Eq. 5) from the stimulus closest to ½ of the RF size (out of all tested sizes) to the RF-sized stimulus. Column 3: change in FF from the RF-sized stimulus to the large stimulus (closest to 2 × RF size). Column 4: the p value for the second column. Column 5: FF change from stimuli smaller to larger than RF size. Sizes are selected to match the mean spike count across neurons (spike count change <3%, p > 0.05, for all experiments). Column 6: p value for column 5. The p values were computed with a one-sided paired samples t test of the null hypothesis that the difference between the two conditions had mean ≤ 0.

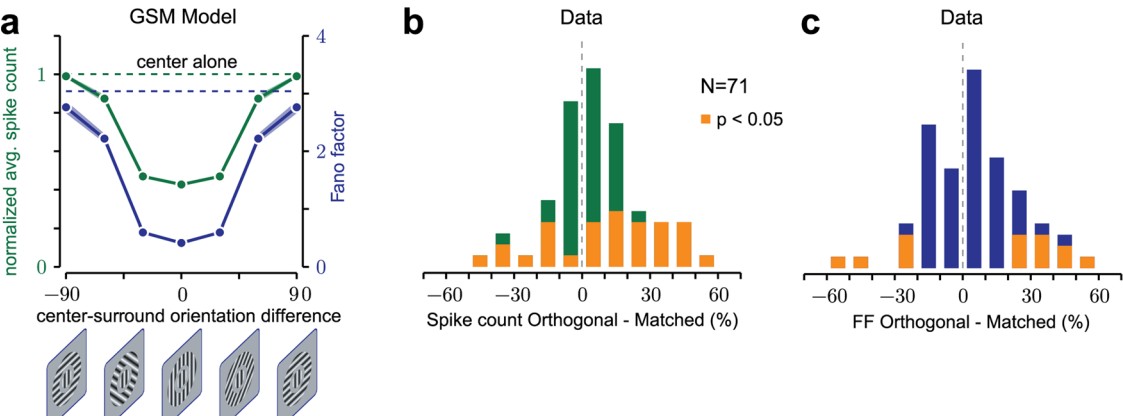

**Fig. 3 Surround reduction of variability is orientation tuned. a** Surround-orientation tuning of the mean spike count (green circles) and FF (blue circles), relative to the center stimulus alone (dashed lines) in the GSM model. Error bars: 68% c.i. computed by bootstrapping. Scaling factor (Methods Eq. 4) $c = 40$ (**b, c**) Percent change in the mean spike count (**b**) and Fano factor (**c**) from orthogonal to matched surround orientation, in V1 of two awake fixating macaques. Yellow bars denote neurons with a significant change across conditions. The difference is considered significant when the 68% c.i.'s of the two conditions (computed by bootstrapping) do not overlap.

Our analysis shows that surround suppression of variability in V1 is tuned to the orientation of surround stimuli, in a manner similar to the tuning of firing rate suppression, suggesting partly shared mechanisms. In the GSM framework, this tuning arises because only matched surround stimuli provide information about the global modulator and thus reduce uncertainty.

## Discussion

We have presented a theoretical framework that explains V1 variability and its modulation by spatial context in natural images, as reflecting probabilistic inference about local features in visual inputs. Our work builds on the theory of neural sampling[10,20,34,39], in which neuronal variability encodes uncertainty of the inferences, and offers two main contributions. First, we established a precise link between V1 response variability and the statistics of natural images. We showed that the dependence between spike count variance and mean, and the modulation of variability by spatial context are general consequences of probabilistic inference when there are multiplicative interactions between latent variables, which is a widely-adopted description of natural image statistics[27,41,44,60–62]. Second, we validated our model with measurements of V1 activity. Consistent with model predictions, spatial context in images modulated V1 variability beyond the known reduction of variability from spontaneous to stimulus-driven activity[4]. Furthermore, the tuning of contextual modulation of variability was similar to (although weaker than) that of mean spike counts, suggesting shared mechanisms.

**Natural image statistics and contextual modulation of response variability**. Normative models of visual processing have explained properties of V1 representations from optimization and efficiency principles related to the statistics of the natural environment[25–28,63,64]. This work has typically addressed only the trial-averaged spike counts. However, across-trial variability is substantial in cortex and can strongly influence perception[17,38,65,66]. Understanding cortical processing requires addressing this variability, which we have done via the neural sampling theory.

The hypothesis that neuronal variability reflects sampling from a distribution[34] is rooted in machine learning research focused on efficient inference schemes[67]. Past work in neural network modeling has shown how samples might be generated dynamically, and in a manner that is fast enough for accurate inference within short, biologically relevant timescales[68–71].

While past work has addressed the plausibility of neural sampling, we have focused instead on contextual effects, for two important reasons. First, contextual effects disambiguate between two key aspects of neural coding: the strength of the stimulus feature represented by the neuron, and the uncertainty about that feature. This is because stimuli in the RF surround do not directly affect the inputs to the RF, but they can modulate uncertainty. This is different from contrast modulation[10] and other common experimental manipulations (e.g., adding stimulus noise[72,73]), that modulate both the strength of a visual feature and its uncertainty. Second, natural visual inputs have rich statistical structure that extends across the visual field. There is abundant evidence suggesting a relation between spatial structure in images and spatial contextual effects in cortex[27,33,63,74,75]. Contextual modulation of V1 trial-averaged responses has been characterized extensively with artificial stimuli[45–49], and is also prominent for natural inputs[51,76]. Past work using the GSM and its extensions has explained a wide range of those phenomena, as reflecting a computation optimized to the statistics of natural images[27,43,44]. The modeling and experimental results presented here are consistent with this prior work, as we report strong and tuned surround suppression of mean spike counts (Figs. 2d–f, 3). But our findings go beyond this previous work, by establishing a general relation between response variability and natural image statistics (Fig. 1c) and relating surround influences on mean spike counts and on variability (Figs. 2d–f, 3).

Our model could be further extended to account for the fact that contextual modulation is weak or absent for some stimuli, such as when contextual inputs are not informative[33]. Variability reduction by stimulus context should be weaker or absent for such uninformative contextual stimuli, which would be consistent with our observations that, when we used natural images, the level of surround suppression of FF varied substantially across images (Fig. 2c and Supplementary Fig. S4), and that suppression was also weaker for orthogonal grating surrounds (Fig. 3c). Although V1 responses agreed well with model predictions, we observed a quantitative discrepancy between the two: contextual modulation of FF and its tuning were much stronger in the model. This could reflect that, in the model, the main source of uncertainty (particularly for the high-contrast stimuli we used), and therefore variability, is the unknown value of the global modulator. Model response variability is therefore extremely sensitive to contextual stimuli. In V1, there are likely multiple latent sources of uncertainty that could partly mask the effects of our experimental manipulation of spatial context. Addressing this

discrepancy may require considering non-sensory contextual factors such as attention and behavioral state[9,39].

**Influences of divisive normalization on variability and other response statistics.** Our mathematical analysis of the GSM inference shows that response strength and variability are jointly modulated by divisive normalization[77,78]. This is because the mean and variance of the inferred distribution of the local features depend divisively on the inferred value of the global modulator (Methods, Eq. 3), which in turn is obtained by combining the inputs corresponding to all features[44] (Methods Eq. 2). Therefore, our model points to divisive normalization as the key operation for surround modulation of rate and variability. There is abundant indirect evidence that normalization modulates responses beyond firing rate. For instance, stimulus manipulations that engage normalization, such as varying contrast and size[49,77], also modulate variability[3,51,52]. In addition, although the mechanisms of normalization are debated[79], network models based on inhibitory stabilization[80] reproduce many of those stimulus-induced effects, indicating a common mechanism that could control both firing rate[81,82] and variability[82] consistently with normalization.

Other work has established the connection between normalization and variability more directly. A descriptive model of stochastic normalization has been shown to fit changes in variability with stimulus contrast[11] and orientation noise[83], and revealed that, even for fixed stimuli, variability is reduced during epochs of strong normalization[11]. Our analytical results on normalization and variability bridge the gap between this literature and a theory of the computational role of variability.

**Relation to other descriptive models and functional explanations of cortical variability.** Previous work used a GSM to demonstrate stimulus dependent changes in response statistics[10]. In particular, Orbán et al.[10] suggested that a GSM could unify effects of response mean and variability. Our work extends this study in two important aspects. First, Orbán et al. used approximate inference in their GSM, based on a maximum a posteriori estimate for the global scaling variable. Consequently, posterior variance was exclusively due to observation noise, while variance resulting from uncertainty in the global scaling variable was ignored. This required tuning a nonlinear conversion from membrane potential to spike counts to account for realistic response variability[84]. Here, we include both sources of uncertainty—input noise and the unknown global latent variable—and we show that the GSM framework is sufficient to capture the dependence between response mean and variance for a wide range of inputs (Fig. 1c, d), without further tuning the conversion from membrane potential to spikes. Second, the treatment of Orbán et al. was sufficient for a coarse grained account of contextual effects (such as changes in sparseness and reliability), but our analysis unveils a more complex repertoire of contextual effects for natural images, leading to detailed predictions that related statistical dependencies across visual space to contextual modulation of V1 variability.

Another recent model[83] proposes that uncertainty is represented in the response variability, and is thus related to sampling and to our work. However, Hénaff et al.[83] propose that variability is partitioned into two terms, Poisson variability and fluctuations in response gain[7]. Uncertainty is encoded specifically by the amplitude of the gain fluctuations. Different from our work, the Poisson term in that framework does not have a functional role and is left unexplained, and there is no precise relation between V1 variability and the statistics of natural images. In addition, whereas sampling-based representations can

approximate the full posterior distribution, the model of Hénaff et al.[83] focuses only on the mean and variance (uncertainty) of the posterior. Therefore, future experimental work could further distinguish between these theories by comparing higher-order statistics of V1 responses to the corresponding statistics in the visual inputs.

## Methods

### Model of V1 responses

*The Gaussian scale mixture (GSM) generative model.* The observable variables are given by the outputs of linear, oriented filters[54] applied to grayscale input images. We assume oriented filters because they approximate well those optimized to natural images, and also represent a canonical choice for V1 models that used the GSM framework[10,33,43,44,71]. One pair of filters (even and odd phase, forming a quadrature pair) represents the RF of the model neuron, and another eight pairs are uniformly distributed on a circle around the RF, all with the same orientation (represented in Fig. 2a as vertical). The surround filters slightly overlap with the RF filters, to reflect that suppressive surround mechanisms in V1 partly overlap with the RF[49] (see Fig. 2a). The responses of the 18 filters form a 18-dimensional input vector, denoted as **x**.

The generative model uses latent variables to capture the statistics of **x**, as follows:

$$\mathbf{x} = \nu \mathbf{g} + \boldsymbol{\eta}$$
$$\mathbf{g} \sim \mathrm{N}(0, \mathbf{C}_g); \ \nu \sim \mathrm{Rayleigh}(1); \ \boldsymbol{\eta} \sim \mathrm{N}(0, \mathbf{C}_{\mathrm{noise}}) \tag{1}$$

The observable **x** results from the product of the feature vector **g**, which has the same dimensionality of **x**, and a positive scalar $\nu$, that acts as global modulator. The additive noise $\boldsymbol{\eta}$ plays the role of observation noise in the generative model. That is, it accounts for the fact that the GSM is not a perfect model of the statistics of the observable **x** on natural images. As we explain below, this additive noise is also helpful to account for realistic response variability with weak stimuli (Supplementary Fig. S10). We assume that **g** and $\boldsymbol{\eta}$ are generated from multivariate normal distributions, with mean 0 and covariances $\mathbf{C}_g$ and $\mathbf{C}_{\mathrm{noise}}$, respectively; $\nu$ follows a Rayleigh distribution with mean 1. Note that changing the Rayleigh parameter is equivalent to rescaling $\mathbf{C}_g$.

*Model optimization.* The covariance of the noise term, denoted as $\mathbf{C}_{\mathrm{noise}}$ in Eq. 1, is found numerically, by applying the filters to 10,000 white-noise patches. We take the empirical covariance of the resulting outputs and scale it by a free parameter, set heuristically at 0.1 to ensure a realistic response variability for weak inputs (Supplementary Fig. S10). The covariance matrix $\mathbf{C}_g$ is computed by moment matching[85], based on the empirical covariance of filter outputs over 10,000 natural image patches, scaled by a term that accounts for the mixer. This ensures that the model is adapted to natural image statistics, as in previous work[44]. The image patches used for training are considered noise-free, and the noise level in the trained model is tuned heuristically. This choice was motivated by convenience, and by noticing that pixel noise tended to be small, reflecting the digital quality of images and not indicative of sensory noise.

*Probabilistic inference and sampling.* Having defined the generative process, we can express the posterior distribution of the latent feature of interest, for example the center-vertical feature with odd spatial phase, $g_{1+}$, given the filters response $\bar{\mathbf{x}}$ to a test image. This quantity is denoted $P(g_{1+}|\bar{\mathbf{x}})$, and results from an operation of Bayesian inference and marginalization over the other latent variables (Supplementary Text, Section 1). In particular, the global modulator $\nu$ plays a key role in the inference of $g_{1+}$. To gain further insight, we first derived analytical solutions for the regime in which input noise is negligible, i.e., $\boldsymbol{\eta} = \mathbf{0}$. First, $\nu$ can be expressed analytically and approximated for $\lambda \gg 1$ (Supplementary Text, Section 1) as:

$$\mathrm{E}[\nu|\bar{\mathbf{x}}] = \sqrt{\lambda}[1 + O(\lambda^{-1})], \ \text{with} \ \lambda = \sqrt{\sum_{i,j}(\mathbf{C}_g^{-1})_{ij}\bar{x}_i\bar{x}_j} \tag{2}$$

where $O(\lambda^{-1})$ represents a generic function that drops to zero asymptotically with $\lambda^{-1}$. This shows that the estimate of the mixer depends on the outputs of all filters. Second, the distribution of the feature of interest, $P(g_{1+}|\bar{\mathbf{x}})$, can also be expressed in closed-form in the low-noise limit (Supplementary Text, Section 4). Its mean and FF can be approximated as:

$$\mathrm{E}[g_{1+}|\bar{\mathbf{x}}] = \frac{\bar{x}_{1+}}{\sqrt{\lambda}}[1 + O(\lambda^{-1})] \ \text{and} \ \mathrm{FF}[g_{1+}|\bar{\mathbf{x}}] = \frac{\bar{x}_{1+}}{4\lambda\sqrt{\lambda}}[1 + O(\lambda^{-1})] \tag{3}$$

In the approximation above (derived in Supplementary Text, Section 4), the expected value of the feature of interest depends linearly on the input inside the RF, $\bar{x}_{1+}$. However it is scaled by $\sqrt{\lambda}$, a quantity approximately equal to the expected value of the global modulator (Eq. 2), which includes the influence of the surround. The variance instead scales divisively with the square of $\lambda$, which in turn determines the reduction of variability (the FF in Eq. 3) by surround stimulation. This analysis thus shows that, in the GSM inference, divisive normalization influences both the mean and the variance of the posterior distribution, thus providing a

normative explanation for the dependence between spike count variance and mean observed in sensory neurons. Notice also that the expected value and the FF are not always monotonically related, because $\lambda$ depends both on inputs inside and outside the RF, and appears with different exponents in the FF and expected value. For instance, surround stimulation affects only $\lambda$ and thus changes the FF and expected value in the same direction, whereas changing contrast affects both numerator and denominator resulting in opposite scaling of the expected value and FF (Supplementary Fig. S11).

The analytical results in Eq. 3 refer to the reduced model without additive noise. In this formulation, for very small inputs $\bar{x} \approx \mathbf{0}$ the inferred mean and variance converge to zero, resulting in model neurons with an unrealistically silent and stable baseline activity. We therefore extended the generative model to non-zero additive noise, and determined the model neuron responses numerically, by Monte Carlo sampling, implemented through the Stan programming language (https://mc-stan.org/). When comparing the analytical solution for the noiseless model with the simulation results for the full model, we found that, as expected, they differ predominantly in the regime of small inputs, where the model with noise still preserves a non-zero response and variability (Supplementary Fig. S10).

Our choice of a fixed Rayleigh prior for the mixer (in line with past work[33,43,44,86]) is mainly due to mathematical convenience, as it allows us to obtain analytical insights on the scaling of mean and variance with $x_{1+}$ and $\lambda$. Although we focused here on qualitative predictions, for quantitative fits of GSM models to neural data one could leverage the flexibility afforded by modifying the mixer prior and introducing additional free parameters.

*Conversion to spike counts.* For the purpose of our analysis, $x_{1+}$ in Eq. 3 is assumed greater than 0 (e.g., a grating stimulus in-phase with the filter). To cover the general case, and appropriately express neural response and FF in terms of spike counts, we performed the following transformation:

$$r = c\sqrt{g_{1+}^2 + g_{1-}^2} \qquad (4)$$

where $c$ is a fixed parameter set heuristically so that mean responses and FF are in a realistic range (values are reported in the figure captions), and the $\pm$ represent the two spatial phases at the RF position. One strength of this framework (following Orban et al.[10]) is that it is a fully normative model of response variability, and does not need to assume additional noise in the spiking process. We can therefore directly consider the instantaneous response $r$ as a spike count, with a rounding error that is small for sufficiently high $c$. In the no-noise approximation, the mean and variance of $r$ can be expressed analytically, and preserve the behavior of Eq. 4 (see Supplementary Text, Section 5). For the full model, we compute a single-trial response $r$ for each sample of $g_{1+}, g_{1-}$. The mean, variance and FF of the model neuron are then found numerically, using 400 samples.

The simple form of Eq.4 allows for analytical results that provide useful intuitions. However, when testing the GSM response to stimuli of fixed size, we found that an increase in contrast led to a decrease in variance, in conflict with V1 data (Supplementary Fig. S11A, B). This behavior can be easily corrected (Supplementary Fig. S11E, F) by using a different transformation between the latent variable $g$ and the neural response, in the form of a rectified expansive nonlinearity[10]. Note however that the GSM predictions for size tuning and surround-orientation tuning stimuli are qualitatively robust to the specific choice of transformation (Supplementary Fig. S12).

## Neurophysiology

*Animal preparation and data collection.* We recorded data from male adult macaque monkeys (*Macaca fascicularis*), either anesthetized (three animals) or awake (two animals). The protocol and general methods employed for the anesthetized experiments have been described previously[87]. In short, anesthesia was induced with ketamine (10 mg/kg of body weight) and maintained during surgery with isoflurane (1.5–2.5% in 95% O2), switching to sufentanil (6–18 μg/kg per h, adjusted as needed) during recordings. Eye movements were reduced using vecuronium bromide (0.15 mg/kg per h). Temperature was maintained in the 36–37 C° range, and relevant vital signs (EEG, ECG, blood pressure, end-tidal PCO2, temperature, and airway pressure) were monitored continuously to ensure sufficient level of anesthesia and well-being. We implanted a $10 \times 10$ multielectrode array (400 μm spacing, 1 mm length) in V1.

For awake experiments the animal was first familiarized with a restraining chair (Crist Instruments). Then a titanium headpost was implanted under full isoflurane anesthesia in an aseptic environment. Postoperative analgesic (buprenorphine) and antibiotic (enrofloxacin) were provided. After a 6 week recovery period, the animal was trained to fixate in a $1.3° \times 1.3°$ window. Eye position was monitored with a high-speed infrared camera (Eyelink, 1000 Hz). Once sufficient performance was reached, a second surgery was performed in which a craniotomy and durotomy were performed over the occipital cortex. A 96-channel and a 48-channel microelectrode array were implanted in V1 (and a third, 48-channel array in V4, not considered here). The dura was sutured over the arrays and covered with a gelatin film (Duragen). The craniotomy was covered with titanium mesh, held in place with titanium screws. On the first day of recording we mapped the spatial receptive fields of the sampled neurons by presenting small patches of drifting full contrast gratings (0.5° diameter; 4 orientations, 1 cycle/deg, 3 Hz drift rate, 250 ms

presentation) at 25 distinct positions spanning a $3° \times 3°$ region of visual space. Subsequent stimuli were centered in the aggregate RF of the recorded units.

All procedures were approved by the Albert Einstein College of Medicine and followed the guidelines in the United States Public Health Service Guide for the Care and Use of Laboratory Animals.

*Visual stimuli.* Visual stimuli were generated with custom software (EXPO V1.5; https://sites.google.com/a/nyu.edu/expo) and displayed on a cathode ray tube monitor (Hewlett Packard p1230; $1024 \times 768$ pixels, with ~40 cd/m2 mean luminance and 100 Hz frame rate) viewed at a distance of 110 cm (for anesthetized) or 60 cm (for awake). In each session, stimuli were randomly interleaved, separated by a uniform gray screen (blank stimulus). All grating stimuli were presented at 100% contrast.

*Surround modulation experiments.* We measured surround modulation in anesthetized animals with grayscale natural images (as described in[33]). Briefly, we presented 270 images in total, each at two sizes (1° and 3.1–6.7°). These included 90 distinct images. For images with a dominant orientation, we presented four variants rotated in steps of 45°, to increase the probability that each variant would drive at least some of the recorded neurons. Images were presented for 200 ms followed by 100 ms blank screen in pseudo-random order, each repeated 20 times.

*Size-tuning experiments.* We measured size tuning with grayscale natural images, and both static and drifting gratings (Table 1 and Supplementary Fig. S6). In each session of the awake experiments we presented ten natural images (a subset of the 270 described above) masked by a circular window with diameters of 0.34, 0.55, 0.90, 2.4, and 3.8°, with stimulus duration 200 ms and interstimulus interval of 100 ms. Images were presented 60–74 times each. We chose images that evoked strong average responses in a majority of the neurons reported in Coen-Cagli et al.[33]. In separate sessions, we measured size tuning with static circular gratings, with diameters of 0.34, 0.55, 0.90, 2.4, and 3.8°; orientations of 0, 45, 90, and 135°; duration of 250 ms, and interstimulus duration of 100 ms. We set the spatial frequency (1 cycle/deg) to be appropriate for V1 neurons at the recorded eccentricity. Each stimulus was repeated 114–124 times. In the anesthetized experiments, we measured size tuning with static circular gratings, testing a larger range of conditions (diameters of 0.34, 0.55, 0.90, 1.5, 2.4, 3.8, and 6.2°; orientations of 0, 45, 90, and 135°), and repeated each stimulus 20 times.

*Surround-orientation tuning experiments.* We measured orientation tuning of surround modulation in two awake monkeys, using static compound gratings with a spatial frequency of 1 cyc/deg presented for 200 ms (100 ms interstimulus interval). For monkey M we used a central grating of diameter 1°, orientations of 0 and 90°, and an annular surround with inner diameter of 1° and outer diameter of 6°, with orientation either matched or orthogonal to the center. For monkey C, the central grating was 0.5° in diameter; orientations were 0, 45, 90, and 135°; and a surround with inner diameter of 1.5° and outer diameter of 5°, with orientation either matched or orthogonal to the center. We introduced this gap between center and surround stimulus, to reduce the extent to which the surround stimulus encroached on the neurons' RFs. The results for both monkeys were qualitatively similar. Therefore, we combined them in our analyses.

**Data analysis.** For each electrode, we extracted waveform signals (sampled at 30 kHz) whenever the extracellular voltage exceeded a threshold of 5 times the square root of the mean square signal on each channel. We then sorted waveforms manually using Plexon Offline Sorter V3, and isolated both single and multi-unit clusters, here both referred to as neurons. Data analysis was then performed in Julia 1.5 (https://julialang.org).

*Characterization of neuronal responses and inclusion criteria.* We computed spike counts in a fixed window with length equal to the stimulus duration, shifted by 50 ms after stimulus onset. We also computed baseline activity in the 50 ms window from 20 ms before to 30 ms after stimulus onset. We excluded from further analyses all neurons that were not driven by any stimulus above baseline + 1 std. We also excluded all natural images and grating orientations that, when presented at a size closest to 1° (out of those presented), did not drive the neurons above the baseline + 1 std. Next, we defined the response latency of each neuron as the first time at which the peristimulus time histograms (regularized using a smoothing cubic spline with parameter $2 \cdot 10^{-6}$) at the preferred stimulus size (for size-tuning experiments) or at the smallest size presented (0.5 or 1°, for the other experiments) crossed a threshold of baseline + 1 std. All further analyses were performed on spike counts in windows shifted by the latency of each individual neuron. In the surround modulation experiments on anesthetized monkeys with natural image patches (Fig. 2c) we selected only neurons that responded significantly to at least ten distinct images.

We computed the mean spike count by averaging across trials, and characterized variability by the FF, the ratio between across-trial variance and mean of the spike count. We focused on the FF because, when compared across conditions, it quantifies changes in variability beyond the changes in mean activity. We excluded neurons whose average FF across all stimulus conditions was larger than 2.

Because we were interested in surround modulation of variability, we excluded neurons with RFs not well centered on the stimuli. In experiments with anesthetized animals, we measured multi-unit spatial RFs using small circular oriented gratings (size 0.5°, 4 orientations, 250 ms presentation), fitting the spike counts with a two-dimensional Gaussian. We only kept for further analysis those neurons whose RF center was within 0.4° of the stimulus center. Due to the limited duration of the awake sessions, we could not measure spatial RFs prior to each session. We therefore relied only on the responsivity to small stimuli (described above), and on the following additional criteria (for size-tuning experiments, Fig. 2f and Table 1), as a proxy for appropriate stimulus centering. First, we excluded the neurons that had maximum response for very small (0.3°) or very large (>4°) stimuli, because this was indicative of poor centering. Second, we excluded natural images that elicited weak surround suppression of the mean spike count (below 15%). We verified that our results did not change qualitatively when we changed this threshold (Supplementary Fig. S9).

Lastly, we excluded the neurons whose mean spike count was zero for any given stimulus size (for size-tuning experiments) or surround condition (for surround-orientation tuning experiments), because the FF is not defined in those cases. For the surround-orientation tuning experiments (Fig. 3) we analyzed only the preferred orientation out of those presented, to ensure responses were robust enough that we could measure surround suppression effects reliably.

*Statistical analysis*. In the size-tuning experiments (Fig. 2e, f and Table 1) we first computed mean spike count and FF for each neuron, each stimulus size and condition (natural image identity or grating orientation). We then averaged across conditions, using mean for spike counts and geometric mean for FFs, obtaining an area-summation curve for both spike count and FF for each neuron (e.g., Fig. 2e). The differences in FF across sizes were measured as:

$$\%\text{change in FF} = 100 \cdot (FF_\alpha - FF_\beta)/((FF_\alpha + FF_\beta)/2) \tag{5}$$

Where $\alpha$ refers to the stimulus size closer to RF and $\beta$ to the stimulus size approximately twice the RF. To visualize population averages in Fig. 2f and Supplementary Figs. S6, S7, we expressed stimulus size relative to RF size, and then averaged across neurons for each relative size. Note that some of the relative sizes were available only in a subpopulation with a specific RF size. In those points, averages refer to the available neurons. Supplementary Fig. S7 shows instead the groups as separate. For the surround-orientation tuning experiments, we quantified differences in FF also by Eq. 5, but with $\alpha$ representing the stimulus with orthogonal surround, and $\beta$ the stimulus with matching surround. Confidence intervals in the population plots were estimated by bootstrapping.

For the mean-matching tests in the area-summation experiment (Table 1 and Supplementary Fig. S3), we compared the FF between stimuli that were smaller versus larger than the RF, and elicited a similar trial-averaged spike count. Specifically, we pooled the mean spike counts of all neurons and stimuli smaller than the RF in one group, and all neurons and stimuli larger than the RF in a second group. We then subsampled the same number of cases from each group, so as to obtain identical histograms of mean spike counts. Lastly, we compared the FF distributions of the two groups. The $p$ values in Table 1 were computed using a paired sample, one-sided $t$ test of the null hypothesis that differences between samples from the two conditions (i.e., RF size versus $2 \times$ RF size) had mean $\leq 0$.

**Reporting summary**. Further information on research design is available in the Nature Research Reporting Summary linked to this article.

## Data availability
Data for Fig. 2c and Supplementary Fig. S6B are publicly available on the CRCNS data sharing site crcns.org. Data for the other figures can be found at https://doi.org/10.5281/zenodo.4710066 [88]. Natural images used to train the GSM models are publicly available in the BSDS500 database. Source data are provided with this paper.

## Code availability
Code for model simulations and data analysis is available without restrictions on GitHub https://github.com/rubencoencagli/festa-et-al-2020. https://doi.org/10.5281/zenodo.4710150.

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

## Acknowledgements

We thank the R.C.C. and A.K. labs for helpful discussion, the A.K. lab for help with experiments, and Gergo Orbán for discussion and comments on a previous version of the paper. This work was supported by NIH grants EY030578 and EY021371.

## Author contributions

D.F., A.K. and R.C.C. designed the project; D.F. and R.C.C. developed the theory and models; A.A. and A.K. performed the experiments; D.F. and A.D. analyzed the data; D.F., A.K. and R.C.C. wrote the paper.

## Competing interests

The authors have no competing interests.
