## [Peer Review File · Nature Communications]

Reviewers' Comments:

Reviewer #1:

Remarks to the Author:

Using a combination of modelling and experimental data analysis, this paper shows how the variability of neural responses in V1 can be interpreted functionally as representing uncertainty in probabilistic inferences. As such, it extends previous work on neural sampling (particularly by Orban et al 2016, duly cited in the paper) in two significant ways. 1. The authors show that Poisson-like variability is a fundamental consequence of neural sampling under an internal (generative) model assuming a multiplicative interaction between latent causes. 2. The authors use surround-suppression to dissociate changes in the mean and variability of responses, and show that neural sampling predicts both correctly (albeit sometimes only qualitatively, at least in the form implemented in the paper). The paper is well written, and the work is methodologically solid.

I have one main question regarding the argument underlying the "universality" of Poisson-like variability (which is supposed to be one of the main results of the paper). I am simply confused what exactly the authors mean by it. In my book at least (and I believe, in line with the definition used in Ma et al, 2006), the term "Poisson-like variability" means that across some set of conditions that differ in the mean and variance of responses, the Fano factor (FF, the ratio of the variance and the mean) remains constant. In contrast, 2 out of 3 main figures are all about how the FF changes across conditions. This is consistent with Orban et al 2016, in that one of the very hallmarks of sampling is that variability is *not* Poisson-like in the sense defined above.

Moreover, the mathematical derivations (for the simplified noiseless variant of the model) suggest that constant FF will be difficult to achieve in general. According to the derivations, the GSM posterior mean should scale linearly with RF activation (x_1) and inversely with the square root of the global signal (λ), $x_1/(\lambda^{1/2})$, while the variance should scale quadratically with x_1 and inverse quadratically with λ , $x_1^2/(\lambda^2)$ and so the FF should scale linearly with x_1 and inversely with the $\lambda^{3/2}$, $x_1/(\lambda^{3/2})$. Thus, for the FF to be constant, this would require a specific relationship between x_1 and λ to hold across these conditions: x_1 should scale with $\lambda^{3/2}$. It is not entirely obvious to me why this will necessarily be the case. For example, when conditions differ in contrast, I would expect x_1 to scale with λ (not $\lambda^{3/2}$) — because both are scaled by the contrast in fundamentally the same way —, so the FF to decrease. This is indeed the case experimentally — so perhaps worth a mention in the paper, too (though this by itself has been shown by Orban et al, 2016) — but doesn't really qualify as Poisson-like. When conditions differ in stimulus orientation, things depend on how the surround is oriented relative to the RF (so it's more difficult to predict based on mental simulations — i.e. take the following with a grain of salt). If they have similar orientation tuning (which is the scenario mostly considered by the authors), then one would expect again a decreased FF as the mean increases (here, at the preferred orientation). Again, this seems to be the case experimentally (Hennequin et al, Neuron 2018) — but again, not exactly Poisson-like then. If they have similar tuning, but with the surround more weakly tuned, then things might just about balance so that λ changes less than x_1 , and so the FF happens to be roughly constant, i.e. Poisson like — but this then seems to be a bit of a corner case. (If the surround is tuned orthogonally to the RF then the FF should again be modulated, but such that it is maximal when the mean peaks, i.e. at the preferred orientation — unlike in the experiments I know of.)

To me, the intuition you give on p.4 (l.5-19) for these derivations is similarly misleading, as it would suggest that mean scales as $1/\nu$, while variance as $1/\nu^2$, in which case FF would scale as $1/\nu$, and thus not be constant.

I appreciate that Fig. 1D shows behaviour that would phenomenologically pass as "Poisson-like", but then I suspect this is because there were some constraints on the magnitude of variation in x_1 and/or λ , or on their correlation. But then, if this is true, the "Poisson-likeness" is not simply due

to a multiplicative interaction, but also due to some further constraints on natural image statistics which would need to be unpacked.

So, based on these, I suggest the following:

- Abandon references to "Poisson-like" variability or at least make it clear upfront what you mean by it (i.e. a definition that is meaningful in the context of your own results, and preferably compatible with previous usage).
- Dig a little deeper into what guarantees "Poisson-like" behaviour in general (as in Fig 1D) despite your mathematical analysis suggesting that it needs rather special requirements to hold.
- Before testing more nuanced (and for your purposes, undoubtedly more directly relevant) effects of the stimulus on variability, why not start with some more basic effects as a sanity check, and see how variability changes with contrast and stimulus tuning. There is data available for both in previous papers (Orban et al, 2016; Hennequin et al, 2018) — as well as in your own recordings, I am sure.

Minor:

- Orban et al, 2016 also looked at a much more simpler form of the effect of stimulus size on variability (their Fig.5) — which, reassuringly, is consistent with your findings. This is perhaps worth mentioning in the Discussion.
- p.2, l.35: I couldn't find the Scott et al 2014 reference in the bibliography.
- p.7, l.5: brown and blue seem to be swapped. I didn't understand the "The surround stimulus does not change the joint distribution of g_c and ν " statement, as the figure seems to clearly show that the marginals change (in line with the second half of the same sentence which says that the "estimates" for g_c and ν change), so the joint must also change. I assume you meant that the constraint coupling g_c and ν does not change — but then you should say that to avoid confusion about the joint.
- p.11, l.9: Echeveste et al, Nature Neuroscience in press (accessible as a 2019 preprint on bioRxiv) may be a relevant reference for networks of neurons doing sampling. Btw, neither that paper, nor the Hennequin et al, 2014 paper you already cite use spiking neurons, but I guess that's also not the main thing you want to emphasise here.
- p.12, l.1-17: When discussing mechanisms of divisive normalisation that may be responsible for changes in both response means and variability, perhaps you want to mention Hennequin et al, Neuron 2018 (building on Ahmadian et al. 2013, and Rubin et al. 2015) which shows that the same SSN network mechanism does indeed both.
- p.13, l.5: Orban et al, 2016 did not learn the filters either, only the prior covariance (and observation noise).
- p.13, l.16: It is a bit dissatisfying to justify observation noise in the generative model post-hoc, by recourse to neural responses once it's inverted and interpreted via sampling. I would think observation noise can be justified in general by model mismatch: the simple generative model of the GSM is clearly mismatched to true image statistics and this observation noise is a poor man's way of acknowledging this mismatch in the model in a statistically (semi-)principled way.
- p.13, l.21-27: The model optimization procedure seems intuitive but it's not clear why you didn't try to learn your parameters "properly" by maximum likelihood. That would help you avoid heuristics (scaling of C_{noise}) and statistical inconsistencies of the sort that you seem to have now: in the

generative model you assume that images are a sum of signal (ν times g) and noise (η) but then you fit the prior covariance of g as if images were pure signal.

- p.13, l.30: I found it a little misleading that you call \tilde{x} "test image" (as if x lived in pixel space). I would say "filter outputs for a test image".

- p.14, l.5, Eq.3: I appreciate that you are building things up here, but referring to the FF of a quantity (g) that can go negative seems rather odd. At least you should acknowledge this here, and you could say that this is an interim result that will be inherited by the quantity that corresponds to actual firing rates. Btw, even firing rates are not quite the quantities for which FF should be measured, they should be spike counts. (And, somewhat sloppily, you use "firing rate" and "spike count" interchangeably when referring to the model, even in the subsection supposed to define these quantities, p.14, l.21-30.) In fact, the way you measure FF for firing rates implicitly assumes a spiking process in which there is no additional variability, so that all the spike count variability is due to variability in firing rates. This is in contrast with standard LNP or GLM or SRM-based approaches (and in line with Orban et al, 2016) that assume Poisson spiking so it's probably worth pointing out explicitly somewhere in the text.

- p.15, l.1: I am sure you didn't mean to say that you were implanted.

- Fig. S4: x-axis labels need an extra space between "relative" and "to".

- Fig. S6: Please explain in the "Methods" section of the caption why you used a special 2D GSM here instead of the usual GSM model used elsewhere in the paper.

- SM, p. 13, Eq. S25: A factor of 4 seems to be missing from the denominator if the FF.

I enjoyed reading your paper.

Mate Lengyel

Reviewer #2:

Remarks to the Author:

This paper from Festa et al examines the theoretical foundation for variability in neuron firing rates. Specifically, they propose that characteristics of neural variability can be explained by probabilistic inference in a model (the GSM) tuned to natural image statistics. The authors show, through analytic arguments, that this model generates empirical Poisson-like variability (mean-scaled variance). Furthermore, the model makes specific predictions about the effect of visual spatial context on spiking variability that the authors then validate in empirical monkey V1 data.

This is an interesting and detailed paper that proposes a specific process - perceptual inference using a GSM generative model - to explain both existing and novel features of neuronal variability. While there is a lot of technical material for a general reader not well-versed in computational neuroscience, the writing is clear and the concepts generally well-presented. The variability of neural activity is a relatively unexplored feature of the brain, and the examination of *how* such variability might arise, and what purpose it may serve, is an important topic. I think this paper is a nice extension of the authors' previous theoretical and empirical work; however, there are a couple of issues I would like to see addressed in a revision, especially the dependence of the results on natural image statistics.

Main points

(1) Linking the GSM to natural image statistics.

A major conclusion is that this paper establishes “a general relation between response variability and natural image statistics” (pg 11, line 29); given this, I think the manuscript can better convey the necessity of a statistics-dependent process to the variability findings.

First, as a narrative issue, the link between the GSM and natural image statistics is not entirely clear. The first set of results (pg 3-5) emphasize the performance of the GSM, including analytic derivations and simulations of inference on real images. However, the text never states clearly that the GSM - and in particular the covariance matrix for the features - is trained to natural images; this is stated in the Methods, but it is an important point to make in the Results text for the general reader. On a related note, the role of statistics is unclear in the schematic depiction of the inference/sampling process (Fig. 1B) - is there any way to clarify this in the graphics?

Second, on a more conceptual level, it is not yet clear to me whether, or to what degree, the GSM variability results relies on being trained to natural statistics. The authors show that the GSM has a number of features, including Poisson-like variability (mean-scaled variance) and mean-dependent FFs. Intuitively, these variability effects arise due to the multiplicative relationship between the inferred mixer and features, but it is not clear that these effects rely on optimizing the GSM to natural images. In other words, do these effects require the covariance matrix (C_g) to be trained to natural image patches? equivalently, would similar results arise if the GSM were trained on artificial stimuli (white noise, gratings, etc.)?

Finally, does the GSM model - trained on natural images - predict different variability effects for gratings vs. natural images? Naively, inference based on natural images should, on average, make different predictions (e.g. about mixer strength) for natural images vs. gratings. There is a hint of this in the empirical data (Table 1, Expt 1 vs 2), with a stronger size-dependent change in FF for gratings. What does the model predict?

(2) Varying sources of empirical V1 data.

A real strength of the paper is robust examination of predicted model results in empirical monkey V1 data, but it is a bit confusing that the sources of data vary across sections. For example: surround stim reduces variability (natural images in 2 anesthetized monkeys), RF vs surround stim effects (natural images in 1 awake monkey; also gratings in 1 (or 2?) awake monkeys and gratings in 3 anesthetized monkeys), orientation selectivity (compound static gratings in 2 awake monkeys).

It would be helpful if the authors clarified why they collected data in both anesthetized and awake preps, and addressed whether the monkey's level of awareness might be related to differences in data; specifically, it appears that size-dependent grating-driven suppression is larger for awake than anesthetized animals (Expt 3 vs Expt 2, table 1). Could this arise because, in the awake state, other factors can contribute to either uncertainty and/or mixer inference (e.g. attention)?

(3) Presentation of model FF effects.

There are a few places where model predictions concerning mean/variance relationships could be better conveyed. The authors state in the text their modeling results re: FF and mean response strength (pg 4, lines 31-33), but the model predictions are never actually quantified or shown as a figure. I think it would be helpful if the authors could add a figure panel - for example as Fig 1E - depicting model variance:mean characteristics as mean response changes. Also, in terms of the relationship between FF and surround stimulation, the authors state that in their empirical data “the vast majority (91.3%) had a lower FF for large images than small ones, consistent with model predictions” (pg 6 lines 24-25) - however, the paper doesn't show model predictions for FF and surround stimulation.

Other points

pg 4, line 2 - typo: "known as Poisson-like response statistics; known as Poisson-like response statistics"^{SEP}

pg 8, line 30 - "we also observed the same trends in data" I'd suggest using a different word than "trends", since it has statistical connotations (specifically borderline significance) whereas the results are very significant

pg 8 - It is unclear how many monkeys contribute to the awake grating data. The text says "we measured responses to static gratings both in the same animal and in a second awake monkey, and similar results" but the 19 neurons in Expt 2 are described in the Fig S4A legend as "one monkey".

table 1 - The column headings with "FF change" are unclear since the term doesn't specify a direction of change, would be better as "decrease". Also, "vs." doesn't convey which is the comparison condition. The table legend clarifies it, but it would be easier if the headings themselves were clearer.

supp, line 56 - typo "ad FFs"

We are glad the reviewers found our work a timely and valuable contribution, and we thank them for their very useful, constructive and detailed comments. We first summarize the main changes to the manuscript, and then provide detailed, point-by-point replies and an indication of what changes were made to the manuscript to address them. In our replies, the changes to the manuscript are referenced by page and line numbers. In the manuscript, they are highlighted in yellow.

We have clarified our main points that did not come out clearly in the first version, and we have conducted substantial new model simulations and analyses to address the reviewers comments. First, regarding our definition of Poisson-like variability, we no longer use the term because it was used in a manner not consistent with past literature, and so it was confusing. Instead, we refer to a dependence between mean and variance of the neural responses, and we have added a new Fig. 1E to show that the Fano factor in our model is not constant across stimuli, consistent with data and in agreement with R1's comments. Second, regarding the relation to natural image statistics pointed out by R2, we now clarify that both the structure of our model (multiplicative interactions) and the tuning of its parameters are important to capture image statistics, but the structure is more important than fine-tuning the parameters to capture V1 single-neuron responses. We demonstrate this with new simulations and model comparisons in Supplementary Figures 11 and 12, and an additional reviewer figure, for models with different structures (additive versus multiplicative) and trained on different image sets (natural, gratings, noise). Third, following R1's suggestions, we have also included new Supplementary Figures 9 and 10, to illustrate the model behavior for simpler stimuli (orientation and contrast tuning) and compare it more directly to the implementation of Orban et al 2016. Lastly, we have extended the simulations of Fig. 1D, and also modified the text, to more clearly indicate the comparison between our model and our data, as requested by R2. All other minor points have been addressed as well.

Reviewer #1 (Remarks to the Author):

Using a combination of modelling and experimental data analysis, this paper shows how the variability of neural responses in V1 can be interpreted functionally as representing uncertainty in probabilistic inferences. As such, it extends previous work on neural sampling (particularly by Orban et al 2016, duly cited in the paper) in two significant ways. 1. The authors show that Poisson-like variability is a fundamental consequence of neural sampling under an internal (generative) model assuming a multiplicative interaction between latent causes. 2. The authors use surround-suppression to dissociate changes in the mean and variability of responses, and show that neural sampling predicts both correctly (albeit sometimes only qualitatively, at least in the form implemented in the paper). The paper is well written, and the work is methodologically solid.

Thank you for the positive evaluation!

R1.1 *I have one main question regarding the argument underlying the "universality" of Poisson-like variability (which is supposed to be one of the main results of the paper). I am simply confused what exactly the authors mean by it. In my book at least (and I believe, in line with the definition used in Ma et al, 2006), the term "Poisson-like variability" means that across some set of conditions that differ in the mean and variance of responses, the Fano factor (FF, the ratio of the variance and the mean) remains constant. In contrast, 2 out of 3 main figures are all about how the FF changes across conditions. This is consistent with Orban et al 2016, in that one of the very hallmarks of sampling is that variability is *not* Poisson-like in the sense defined above.*

We agree that our use of Poisson-like in the original manuscript was unconventional, different from Ma et al 2006, and could lead to confusion. Thank you for pointing this out. We used "Poisson-like" loosely, to indicate that the variance is roughly correlated with the mean response, across stimuli/conditions. We did not mean that the Fano factor (FF) is constant across stimuli/conditions. As you correctly point out, the FF in our model (as in

Orban et al 2016) is not constant in general, and one of our key results is that the FF in V1 is modulated by the stimulus manipulations we considered, in qualitative agreement with the model predictions.

To remove this source of potential confusion, we have changed “Poisson-like” to “dependence between variance and mean” in the revised manuscript. We have also modified the text to clarify that our finding that multiplicative interactions between latents imply a relation between the mean and variance qualitatively consistent with V1 data—namely, the mean and variance are positively correlated across inputs, but not linearly related (which would imply constant FF).

List of changes

DEL (abstract) > Here we combine analysis of image statistics and recordings in macaque V1 to show that probabilistic inference tuned to natural image statistics explains Poisson-like variability, and the modulation of V1 activity and variability by spatial context in images.

ADD p1.7-10 (abstract) > Here we combine analysis of image statistics and recordings in macaque V1 to show that probabilistic inference tuned to natural image statistics explains the widely observed dependence between spike-count variance and mean, and the modulation of V1 activity and variability by spatial context in images.

DEL > First, we show analytically that Poisson-like variability (CITATIONS...) emerges in the GSM from the multiplicative interactions between local and global image features. Second, we show that stimuli in the RF surround modulate these interactions, and thus also response variability.

ADD p3.17-21> First, we show analytically that the dependence between spike-count variance and mean observed empirically (CITATIONS...) emerges in the GSM from the multiplicative interactions between local and global image features. Second, we show that stimuli in the RF surround modulate these interactions, and thus also response variability.

DEL (title) > Poisson-like variability reflects multiplicative interactions between latent variables

ADD p3.29-30 > The dependence between spike-count variance and mean reflects multiplicative interactions between latent variables

DEL > This example illustrates why a neuron whose responses reflect samples from the inferred distribution of g should display Poisson-like statistics.

ADD p4.18-23 (with additional changes that address R1.4) > This example illustrates why a neuron whose responses reflect samples from the inferred distribution of g should display a dependence between mean and variance in its response statistics. Note that this dependency is not linear, nor do mean and variance strictly follow each other as they would in a Poisson process. In general, the relative scaling depends on model choices, such as the uncertainty on the priors and, for high dimensional inputs, the stimulus structure (as explained in the next section).

DEL > These analyses confirm the intuition that Poisson-like variability in the GSM emerges from the multiplicative interactions between the global modulator and the local variables.

ADD p5.1-3 > These analyses confirm the intuition that the dependence between posterior variance and mean observed in the GSM emerges from the multiplicative interactions between the global modulator and the local variables.

DEL (figure caption title) > Poisson-like variability explained by sampling-based inference in the GSM model.

ADD p5.7-8 > Sampling-based inference in the GSM model explains the dependence between spike-count variance and mean.

DEL > First, we established a precise link between V1 response variability and the statistics of natural images. We showed that Poisson-like variability and its modulation by spatial context is a general consequence of probabilistic inference when there are multiplicative interactions between latent variables, which is a widely-adopted description of natural image statistics.

ADD p10.13-19 > First, we established a precise link between V1 response variability and the statistics of natural images. We showed that the dependence between spike-count variance and mean, and the modulation of variability by spatial context are general consequences of probabilistic inference when there are multiplicative interactions between latent variables, which is a widely-adopted description of natural image statistics.

DEL > ...and we show that the GSM framework is sufficient to capture Poisson-like responses for a wide range of inputs (Fig. 1C,D)...

ADD p12.31-32 >... and we show that the GSM framework is sufficient to capture the dependence between response mean and variance for a wide range of inputs (Fig. 1C,D)...

DEL > This analysis thus shows that, in the GSM inference, divisive normalization influences both the mean and the variance of the posterior distribution, thus providing a normative explanation for Poisson-like variability.

ADD p14.17-20> This analysis thus shows that, in the GSM inference, divisive normalization influences both the mean and the variance of the posterior distribution, thus providing a normative explanation for the dependence between spike-count variance and mean observed in sensory neurons.

DEL *p4.2 of previous draft* > known as Poisson-like response statistics

DEL > This required tuning the nonlinear conversion from membrane potential to firing rate to account for Poisson-like variability (Carandini 2004).

ADD p12.28-30 > This required tuning a nonlinear conversion from membrane potential to spike counts to account for realistic response variability (Carandini 2004).

R1.2 *Moreover, the mathematical derivations (for the simplified noiseless variant of the model) suggest that constant FF will be difficult to achieve in general. According to the derivations, the GSM posterior mean should scale linearly with RF activation (x_1) and inversely with the square root of the global signal (λ), $x_1/(\lambda^{1/2})$, while the variance should scale quadratically with x_1 and inverse quadratically with λ , $x_1^2/(\lambda^2)$ and so the FF should scale linearly with x_1 and inversely with the $\lambda^{3/2}$, $x_1/(\lambda^{3/2})$. Thus, for the FF to be constant, this would require a specific relationship between x_1 and λ to hold across these conditions: x_1 should scale with $\lambda^{3/2}$. It is not entirely obvious to me why this will necessarily be the case.*

That is correct. As explained also in our response above, the GSM posterior mean and variance do not scale equally in general, and so the FF is not constant. As you point out, this is clear from our analytic results for the noiseless case, and borne out in the simulations of the model with input noise. Our main point is that, because of the multiplicative interaction, the mean and variance are not independent of each other. The changes described in our previous response address this.

R1.3 For example, when conditions differ in contrast, I would expect x_1 to scale with λ (not $\lambda^{3/2}$) — because both are scaled by the contrast in fundamentally the same way —, so the FF to decrease. This is indeed the case experimentally — so perhaps worth a mention in the paper, too (though this by itself has been shown by Orban et al, 2016) — but doesn't really qualify as Poisson-like. When conditions differ in stimulus orientation, things depend on how the surround is oriented relative to the RF (so it's more difficult to predict based on mental simulations — i.e. take the following with a grain of salt). If they have similar orientation tuning (which is the scenario mostly considered by the authors), then one would expect again a decreased FF as the mean increases (here, at the preferred orientation). Again, this seems to be the case experimentally (Hennequin et al, Neuron 2018) — but again, not exactly Poisson-like then. If they have similar tuning, but with the surround more weakly tuned, then things might just about balance so that λ changes less than x_1 , and so the FF happens to be roughly constant, i.e. Poisson like — but this then seems to be a bit of a corner case. (If the surround is tuned orthogonally to the RF then the FF should again be modulated, but such that it is maximal when the mean peaks, i.e. at the preferred orientation — unlike in the experiments I know of.)

We agree with your predictions, and have performed new simulations to test them. The results are described in detail in the response to **R1.8**.

R1.4 To me, the intuition you give on p.4 (l.5-19) for these derivations is similarly misleading, as it would suggest that mean scales as $1/\nu$, while variance as $1/\nu^2$, in which case FF would scale as $1/\nu$, and thus not be constant.

We agree that, because of our confusing usage of “Poisson-like”, the intuitive explanation on p.4 (l.5-19) sounded misleading. Our goal with this text is to illustrate why the variance changes together with the mean, but not to imply that the variance and mean scale equally. For extra clarity, we have now stated explicitly that the scaling of mean and variance is different, so that their ratio is not constant.

See change to p4.16 in R1.1.

R1.5 I appreciate that Fig. 1D shows behaviour that would phenomenologically pass as “Poisson-like”, but then I suspect this is because there were some constraints on the magnitude of variation in x_1 and/or λ , or on their correlation. But then, if this is true, the “Poisson-likeness” is not simply due to a multiplicative interaction, but also due to some further constraints on natural image statistics which would need to be unpacked.

Now that we have clarified what we meant by Poisson-like, we think it is also clear that we didn't mean that constant FF is a general behavior of our model, nor do we claim that Fig. 1D shows constant FF across stimuli. In Fig. 1D (and Fig. S8.A), the roughly linear relation between log-mean and log-variance, with slope larger than 1, implies that FF overall tends to increase with the mean across those images. To make this more explicit, we have also added a new Fig. 1E that shows how FF changes as a function of the mean response. This is not a consequence of specific constraints or choices of model parameters, but rather reflects the range of filters activations elicited by those images (large natural images). However, again, other specific image manipulations, such as contrast, can also produce FF decreasing with the mean – see point **R1.8** – and one could also construct corner cases where the FF remains constant.

Note that in the revised manuscript, we have expanded the simulations of Fig. 1D. For details on how 1D changed, and the caption of 1E, see comments to R2.6.

R1.6 So, based on these, I suggest the following:

- Abandon references to "Poisson-like" variability or at least make it clear upfront what you mean by it (i.e. a definition that is meaningful in the context of your own results, and preferably compatible with previous usage).

Yes, addressed above.

R1.7 - Dig a little deeper into what guarantees "Poisson-like" behaviour in general (as in Fig 1D) despite your mathematical analysis suggesting that it needs rather special requirements to hold.

Addressed in **R1.5**.

R1.8 - Before testing more nuanced (and for your purposes, undoubtedly more directly relevant) effects of the stimulus on variability, why not start with some more basic effects as a sanity check, and see how variability changes with contrast and stimulus tuning. There is data available for both in previous papers (Orban et al, 2016; Hennequin et al, 2018) — as well as in your own recordings, I am sure.

Thank you for the suggestion. We have performed new model simulations for contrast and orientation tuning. In agreement with the analytic results of the noiseless GSM and with your predictions (**R1.3**), in our main implementation of the GSM the FF decreases a) as contrast increases (new Supplementary Fig. S9A-B) and b) as stimulus orientation becomes more similar to the neuron's preference (new Supplementary Fig. S9C-D).

One caveat is that, in our implementation, increasing the contrast of a RF-sized stimulus leads to a net reduction of response variance (a small reduction, due to the term $O(\lambda^{-1})$ in eq. S24), which is in contrast to V1 data (new Supplementary Fig. S9B). This behavior can be easily corrected by using a rectified expansive nonlinear (exponential) transformation between the latent variable g and the neural response, as in Orban et al 2016. To verify this, we repeated the simulations with the following nonlinearity: $r = \alpha \lfloor g_{1+} + \beta \rfloor_+^\gamma$

where $\lfloor \cdot \rfloor_+$ indicates a rectified linear function., With this nonlinearity, we obtained qualitatively similar results for the FF (it decreases with contrast, Supplementary Fig. S9E-F, and with stimulus orientation difference, Supplementary Fig. S9G-H), while also obtaining a net increase of the variance with contrast.

This prompted us to compare the results of our model without (as in our manuscript) and with expansive nonlinearity, for the main simulations of interest: size tuning and orientation tuning of the surround. We find that the results are qualitatively the same (Supplementary Fig. S10), and therefore the results reported in the manuscript are not dependent on our specific choice of the transformation from g to neural activity.

We have decided to keep the model without expansive nonlinearity as our primary choice for the manuscript, because it allows for simpler analytical results that provide useful intuitions. However, we now point out in the Methods that it has a limitation for contrast tuning as described above, and that this limitation can be addressed by using an expansive nonlinearity.

List of changes :

ADD p15.11-17 > The simple form of Eq.4 allows for analytical results that provide useful intuitions. However, when testing the GSM response to stimuli of fixed size, we found that an increase in contrast led to a decrease in variance, in conflict with V1 data (Supplementary Fig. S9A-B). This behavior can be easily corrected (Supplementary Fig. S9E-F) by using a different transformation between the latent variable g and the neural response, in the form of a rectified expansive nonlinearity (Orban et al 2016). Note however that the GSM predictions for size tuning and surround-orientation tuning stimuli are qualitatively robust to the specific choice of transformation (Supplementary Fig.~S10).

Minor:

- Orban et al, 2016 also looked at a much more simpler form of the effect of stimulus size on variability (their Fig.5) — which, reassuringly, is consistent with your findings. This is perhaps worth mentioning in the Discussion.

Yes, we have added this text:

DEL p12.28 > Second, our analytical results addressed more directly the relation...

ADD p12.33-37 > Second, the treatment of Orbán et al was sufficient for a coarse grained account of contextual effects (like changes in sparseness and reliability), but our analysis unveils a more complex repertoire of contextual effects for natural images, leading to detailed predictions that related statistical dependencies across visual space to contextual modulation of V1 variability.

- p.2, l.35: I couldn't find the Scott et al 2014 reference in the bibliography.

This was miscited. The first author is Hunsberger and the full reference is :

Hunsberger, E., M. Scott and C. Eliasmith (2014). "The competing benefits of noise and heterogeneity in neural coding." *Neural Comput* 26(8): 1600-1623.

- p.7, l.5: brown and blue seem to be swapped. I didn't understand the "The surround stimulus does not change the joint distribution of g_c and nu " statement, as the figure seems to clearly show that the marginals change (in line with the second half of the same sentence which says that the "estimates" for g_c and nu change), so the joint must also change. I assume you meant that the constraint coupling g_c and nu does not change — but then you should say that an avoid confusion about the joint.

Thank you. We have modified the text as suggested.

List of changes:

DEL > Noise-free GSM model applied to an image with (blue) and without (brown) surround. The surround stimulus does not change the joint distribution of g_c (the local feature associated with the RF center) and v , but it influences the estimate of the modulator v and therefore also the estimate of g_c .

ADD p7.4-8> Noise-free GSM model applied to an image without (blue) and with (brown) surround. The surround stimulus does not change the constraint between g_c (the local feature associated with the RF center) and nu , but it influences the estimate of the modulator and therefore also the estimate of g_c .

- p.11, l.9: Echeveste et al, *Nature Neuroscience* in press (accessible as a 2019 preprint on bioRxiv) may be a relevant reference for networks of neurons doing sampling. Btw, neither that paper, nor the Hennequin et al, 2014 paper you already cite use spiking neurons, but I guess that's also not the main thing you want to emphasise here.

Thank you for the pointer, we have added this reference in Discussion, and removed 'spiking' as follows:

DEL> Past work has shown how samples might be generated dynamically with networks of spiking neurons, and in a manner that is fast enough for accurate inference within short, biologically relevant timescales (Hennequin, Aitchison et al. 2014, Legenstein and Maass 2014, Savin and Deneve 2014).

ADD p11.6-10> Past work in neural network modeling has shown how samples might be generated dynamically, and in a manner that is fast enough for accurate inference within short, biologically relevant timescales (Echeveste et al. 2020, Hennequin, Aitchison et al. 2014, Legenstein and Maass 2014, Savin and Deneve 2014).

- p.12, l.1-17: *When discussing mechanisms of divisive normalisation that may be responsible for changes in both response means and variability, perhaps you want to mention Hennequin et al, Neuron 2018 (building on Ahmadian et al. 2013, and Rubin et al. 2015) which shows that the same SSN network mechanism does indeed both.*

We have added this reference in Discussion:

ADD p12.11-15 > In addition, although the mechanisms of normalization are debated, network models based on inhibitory stabilization (Ahmadian et al. 2013) reproduce many of those stimulus-induced effects, indicating a common mechanism that could control both firing rate (Rubin et al. 2015; Hennequin et al, Neuron 2018) and variability (Hennequin et al, Neuron 2018) consistently with normalization.

- p.13, l.5: *Orban et al, 2016 did not learn the filters either, only the prior covariance (and observation noise).*

We have corrected the text by removing the sentence “rather than learning this linear transform as in other applications of the GSM (e.g. Orbán, Berkes et al. 2016),”

DEL > Here, rather than learning this linear transform as in other applications of the GSM (e.g. Orbán, Berkes et al. 2016), we assume oriented filters because they approximate well those learned on natural images, and also represent a canonical choice for V1 models.

ADD p13.6-10> We assume oriented filters because they approximate well those optimized to natural images, and also represent a canonical choice for V1 models that used the GSM framework (e.g. Coen-Cagli et al 2009; 2012; 2015; Orbán, Berkes et al. 2016 ; Echeveste et al 2020).

- p.13, l.16: *It is a bit dissatisfying to justify observation noise in the generative model post-hoc, by recourse to neural responses once it's inverted and interpreted via sampling. I would think observation noise can be justified in general by model mismatch: the simple generative model of the GSM is clearly mismatched to true image statistics and this observation noise is a poor man's way of acknowledging this mismatch in the model in a statistically (semi-)principled way.*

We agree. In the modified text, we have added the statistical justification for the observation noise, before the existing text that explains its role in matching neural variability for low inputs. We now write

DEL > The additive noise η is necessary to account for realistic response variability with weak stimuli (Supplementary Fig. S8).

ADD p13.19-22> The additive noise η plays the role of observation noise in the generative model. That is, it accounts for the fact that the GSM is not a perfect model of the statistics of the observable x on natural

images. As we explain below, this additive noise is also helpful to account for realistic response variability with weak stimuli (Supplementary Fig. S8).”

- p.13, l.21-27: *The model optimization procedure seems intuitive but it's not clear why you didn't try to learn your parameters "properly" by maximum likelihood. That would help you avoid heuristics (scaling of C_{noise}) and statistical inconsistencies of the sort that you seem to have now: in the generative model you assume that images are a sum of signal (ν times g) and noise (η) but then you fit the prior covariance of g as if images were pure signal.*

We agree that maximum likelihood would be the ideal approach to fit the model on the input statistics. In early attempts, however, we found that the optimal noise level tended to be small, indicative of the resolution of the digital images used for training, and not of perceptual noise in natural vision. We therefore opted to keep the noise level as a free parameter, and considered digital images noise-free. Once those constraints were set, the moment-matching approach appeared unbiased and effective when tested on synthetic data.

Also, related to the minor point p 13, l. 16 , Gaussian noise did not seem well-suited to make up for statistical mismatches between model and training set. In data-science applications of models of the GSM family, the emphasis is typically on a more flexible mixer prior, to better control the long tails and the high order mean-variance dependencies (Doulgeris and Eltoft, 2009). Similarly, in past work, we have found that hierarchical extensions to Mixtures of GSMs more accurately match natural image statistics (Coen-Cagli et al 2009).

ADD p13.29-35> The covariance matrix C_g is computed by moment-matching (Doulgeris and Eltoft 2009), based on the empirical covariance of filter outputs over 10,000 natural image patches, scaled by a term that accounts for the mixer. This ensures that the model is adapted to natural image statistics, as in previous work (Coen-Cagli, Dayan et al. 2012). The image patches used for training are considered noise-free, and the noise level in the trained model is tuned heuristically. This choice was motivated by convenience, and by noticing that pixel noise tended to be small, reflecting the digital quality of images and not indicative of sensory noise.

- p.13, l.30: *I found it a little misleading that you call \tilde{x} "test image" (as if x lived in pixel space). I would say "filter outputs for a test image".*

Yes, we have changed to “given the filters response \tilde{x} to a test image”.

DEL > ... spatial phase, g_{1+} , given a test image \tilde{x}

ADD p14.1 > spatial phase, g_{1+} , given the filters response \tilde{x} to a test image

- p.14, l.5, Eq.3: *I appreciate that you are building things up here, but referring to the FF of a quantity (g) that can go negative seems rather odd. At least you should acknowledge this here, and you could say that this is an interim result that will be inherited by the quantity that corresponds to actual firing rates. Btw, even firing rates are not quite the quantities for which FF should be measured, they should be spike counts. (And, somewhat sloppily, you use "firing rate" and "spike count" interchangeably when referring to the model, even in the subsection supposed to define these quantities, p.14, l.21-30.) In fact, the way you measure FF for firing rates implicitly assumes a spiking process in which there is no additional variability, so that all the spike count variability is due to variability in firing rates. This is in contrast with standard LNP or GLM or SRM-based approaches (and in line with Orban et al, 2016) that assume Poisson spiking so it's probably worth pointing out explicitly somewhere in the text.*

Thank you for pointing this out. We think one strength of this framework (following Orban et al 2016) is that it is a fully normative model of response variability, which does not need to assume additional noise in the spiking process (see also our response to R1.1). Besides the changes described in R1.1, we now always refer to the model's "spike count" (and reserve "firing rate" for experimental data, where appropriate) as defined in the modified Methods text, as follows:

DEL > In the GSM, the elements of g are not bounded to positive values. In order to simulate a neuronal response (the instantaneous firing rate) we therefore apply the following transformation: [...]

ADD p14.25 – p15.7> For the purpose of our analysis, x_1 in Eq. 3 is assumed greater than 0 (e.g. a grating stimulus in-phase with the filter). To cover the general case, and appropriately express neural response and FF in terms of spike counts, we performed the following transformation: [...] where c is a fixed parameter set heuristically so that mean responses and FF are in a realistic range (values are reported in the figure captions), and the \pm represent the two spatial phases at the RF position. One strength of this framework (following Orban et al 2016) is that it is a fully normative model of response variability, and does not need to assume additional noise in the spiking process. We can therefore directly consider the instantaneous response r as a spike count, with a rounding error that is small for sufficiently high c .

- p. 15, l. 1: *I am sure you didn't mean to say that you were implanted.*

Indeed! Corrected.

DEL > We were implanted...

ADD p15.27-28 > We implanted...

- Fig. S4: *x-axis labels need an extra space between "relative" and "to".*

Corrected.

- Fig. S6: *Please explain in the "Methods" section of the caption why you used a special 2D GSM here instead of the usual GSM model used elsewhere in the paper.*

We have added the following explanation:

DEL S6 label > **Methods:** We generated a 2-dimensional GSM model, with diagonal covariance matrix with elements [5,5], and additive noise that also had a diagonal covariance, with elements [0.01,0.01]. The center input is a sigmoid between 0 and 1, whereas the surround reached 2.5. Mean and FF for the central feature (given the stimulus) were calculated semi-analytically by numerical integration.

ADD S6 label > **Methods:** We opted for a 2D GSM model to remove possible confounders, such as dependencies due to the filter covariance structure. The two dimensions represent, respectively, the encoded latent variable (the center), and the contextual information (the surround). The model parameters C_g and C_{noise} were both diagonal, with elements [5,5] and [0.01,0.01] respectively. The inputs varied parametrically according to size, taking a sigmoidal shape. The center input saturated at 1, whereas the surround input reached a level of 2.5. For simplicity, we considered the input as only positive, and directly identified the spike counts with the latent posterior distribution. Mean and FF of the model response were calculated semi-analytically by numerical integration.

- SM, p. 13, Eq. S25: A factor of 4 seems to be missing from the denominator if the FF.

Corrected.

I enjoyed reading your paper.

Thank you!

Mate Lengyel

Reviewer #2 (Remarks to the Author):

This paper from Festa et al examines the theoretical foundation for variability in neuron firing rates. Specifically, they propose that characteristics of neural variability can be explained by probabilistic inference in a model (the GSM) tuned to natural image statistics. The authors show, through analytic arguments, that this model generates empirical Poisson-like variability (mean-scaled variance). Furthermore, the model makes specific predictions about the effect of visual spatial context on spiking variability that the authors then validate in empirical monkey V1 data.

*This is an interesting and detailed paper that proposes a specific process - perceptual inference using a GSM generative model - to explain both existing and novel features of neuronal variability. While there is a lot of technical material for a general reader not well-versed in computational neuroscience, the writing is clear and the concepts generally well-presented. The variability of neural activity is a relatively unexplored feature of the brain, and the examination of *how* such variability might arise, and what purpose it may serve, is an important topic. I think this paper is a nice extension of the authors' previous theoretical and empirical work; however, there are a couple of issues I would like to see addressed in a revision, especially the dependence of the results on natural image statistics.*

Thank you for the positive evaluation!

Main points

(1) Linking the GSM to natural image statistics.

R2.1 - A major conclusion is that this paper establishes “a general relation between response variability and natural image statistics” (pg 11, line 29); given this, I think the manuscript can better convey the necessity of a statistics-dependent process to the variability findings.

Yes, this is a central idea of the manuscript, and we have attempted to emphasize and clarify it in the revised text, as detailed in our replies below.

R2.2 - First, as a narrative issue, the link between the GSM and natural image statistics is not entirely clear. The first set of results (pg 3-5) emphasize the performance of the GSM, including analytic derivations and

simulations of inference on real images. However, the text never states clearly that the GSM - and in particular the covariance matrix for the features - is trained to natural images; this is stated in the Methods, but it is an important point to make in the Results text for the general reader. On a related note, the role of statistics is unclear in the schematic depiction of the inference/sampling process (Fig. 1B) - is there any way to clarify this in the graphics?

Thank you for pointing out this omission, we have now added in Results that our simulation results use a GSM trained on natural images:

DEL > ... we implemented a GSM with oriented filters (Simoncelli and Freeman 1995) spatially arranged to cover the RF of the model neuron and its surround (Fig. 2A; details in Methods).

ADD p4.28-32> we implemented a GSM with oriented filters (Simoncelli and Freeman 1995) spatially arranged to define both the RF of the model neuron and its surround (Fig. 2A; details in Methods). The model was trained on a large ensemble (N=10,000) of natural image patches extracted from the BSDS500 database, (P. Arbelaez et al., IEEE TPAMI, Vol. 33, No. 5, 2011; <https://github.com/BIDS/BSDS500>)

R2.3 - *Second, on a more conceptual level, it is not yet clear to me whether, or to what degree, the GSM variability results relies on being trained to natural statistics. The authors show that the GSM has a number of features, including Poisson-like variability (mean-scaled variance) and mean-dependent FFs. Intuitively, these variability effects arise due to the multiplicative relationship between the inferred mixer and features, but it is not clear that these effects rely on optimizing the GSM to natural images. In other words, do these effects require the covariance matrix (C_g) to be trained to natural image patches? equivalently, would similar results arise if the GSM were trained on artificial stimuli (white noise, gratings, etc.)?*

We agree this is a key point, and we have performed several new simulations to address it. In short, both the structure of the GSM (multiplicative interactions between local and global latents) and its parameters, contribute to capturing statistical properties of images, but we find that statistics in single neurons. Our statement that this work shows a relation between response variability and natural image statistics refers to the multiplicative structure of the GSM, which captures a general property of natural images.

First, to illustrate explicitly the importance of the multiplicative GSM structure, we constructed an alternative model, in which local and global latents interact additively. We trained the additive model on the same natural images as the GSM. Although inference in the additive model could reproduce the size-tuning curve of average firing rate, the variability was independent of the stimulus and therefore the FF was proportional to the inverse firing rate (new Supplementary Fig. S11). This implies that, in the additive model, the FF is minimal when the stimulus size is equal to the receptive field size, and stimulation of the surround increases the FF, which is opposite to our V1 data and GSM predictions (Fig. 2).

Changes :

DEL > Notice that, if the relation between x and g were linear instead of multiplicative, then changes in the inferred value of v would only change the inferred mean of g , not its variance.

ADD p4.23> Notice too that if the mixer term v were additive instead of multiplicative, then changes in its inferred value would only change the inferred mean of g , not its variance, leading to different predictions (Supplementary Fig. S11).

Second, following your suggestion, we have compared the predictions of GSMs trained on a) natural images; b) uniform gratings ; c) white noise, with a range of orientations and contrast levels (Supplementary Fig. S12, first column). We verified that the learned covariance matrix C_g differs substantially in the three cases (Supplementary Fig. S12, second column). With natural images (Supp. Fig. S12, top row), the structure of C_g reflects statistical properties of natural scenes, e.g. stronger covariance between collinear filters due to the abundance of elongated contours in scenes. With large, uniform gratings (Supp. Fig. S12, middle row), the scale of C_g is largest but there is less structure than for natural images, i.e. it simply reflects the geometric alignment of filters, in terms of their orientation and spatial phase. Finally, with white noise (Supp. Fig. S12, bottom row), C_g simply reflects the overlap between filters (larger covariance for pairs of filters that overlap more) and its scale is small.

Despite these differences in the covariance matrices, we found that all models generated qualitatively similar predictions (Supplementary Fig. S12, third column): in all cases, we observed the typical non-monotonic size-tuning curve of average firing rate, and a monotonic reduction of FF for larger stimuli. These analyses indicate that optimizing the GSM parameters to natural images is not required to match qualitatively the modulation of single-neuron response variability that we observed in V1.

In addition, we would also like to point out that, the optimization of C_g to natural images can lead to testable predictions. Specifically, C_g influences the shape of the inferred joint posterior distribution of multiple local latent variables, particularly for weak inputs in which case the posterior reflects more closely the prior. Therefore, in an extended GSM-based model of neural populations, training on natural images versus white noise would lead to qualitatively different predictions. For instance, pairwise noise correlations (which, in the model, would be influenced by the corresponding elements of C_g) would depend strongly on the geometry of the receptive fields for the GSM trained on natural images, but depend only on the distance between receptive fields for the GSM trained on white noise. This, and related predictions, could be tested in the future with new, targeted experiments, but we believe that they fall outside the scope of this paper. Therefore we have decided not to include them in this manuscript, except for a discussion in the caption of Supplementary Fig. S12.

Changes :

DEL > ... we implemented a GSM with oriented filters (Simoncelli and Freeman 1995) spatially arranged to cover the RF of the model neuron and its surround (Fig. 2A; details in Methods).

ADD p4.28-32> we implemented a GSM with oriented filters (Simoncelli and Freeman 1995) spatially arranged to cover the RF of the model neuron and its surround (Fig. 2A; details in Methods). The model was trained on a large ensemble (N=10,000) of natural image patches extracted from the BSDS500 database, (P. Arbelaez et al., IEEE TPAMI, Vol. 33, No. 5, 2011; <https://github.com/BIDS/BSDS500>).

ADD p4.42-46> Importantly, training a GSM on different image sets, such as white noise, led to different parameter values but qualitatively similar predictions for neural responses (Supplementary Fig S.12), indicating that the mean-variance dependence arises from matching the generative model's structure to image statistics (i.e. multiplicative latent interactions) rather than fine-tuning its parameters.

In summary, in our original manuscript we had explained the implications of the GSM multiplicative structure for the responses of the model neuron, but we did not state clearly that a) the structure of the GSM is also important for the generative model to capture image statistics accurately; and b) for the purpose of single-

neuron responses, the GSM structure is more important than the fine tuning of the parameters. Our statement that our work shows a relation between V1 variability and natural image statistics is based on the fact that the multiplicative structure of the GSM makes it a good model of natural images. Our revised text and supplement now address these points.

R2.4 - Finally, does the GSM model - trained on natural images - predict different variability effects for gratings vs. natural images? Naively, inference based on natural images should, on average, make different predictions (e.g. about mixer strength) for natural images vs. gratings. There is a hint of this in the empirical data (Table 1, Expt 1 vs 2), with a stronger size-dependent change in FF for gratings. What does the model predict?

We appreciate the insightful observation. In the figure below we show size tuning for both natural images (left panel, corresponding to Fig. 1D of main text), and uniform gratings at the preferred orientation (right panel). The contrast level of gratings was tuned to elicit a similar mean response at RF size (see insets). For small grating stimuli, the signal is weak compared to natural stimuli and the inference is dominated by noise: the FF is therefore higher. For larger gratings, the FF decreases more sharply, to values lower than for natural images, in line with your hypothesis.

Although this additional novel prediction is interesting, a thorough validation would require additional data, and quantitative studies on how responses change depending on contrast levels and filter positions and shapes. This is because the weaker suppression of FF in natural images relative to gratings, in our data, might also partly reflect that some natural images do not engage surround modulation strongly (termed “heterogeneous” images in Coen-cagli et al 2015), as explained in Discussion page 11, line 32. For a rigorous validation, one should therefore compare modulation of mean responses and FF between gratings, homogeneous natural images, and heterogeneous natural images. This however would require new experiments, and an extension of the model to the MGSM as in Coen-Cagli et al 2015, which we think falls beyond the scope of this paper. However, if the reviewer or editor think we should nevertheless add these results to our supplementary material, we would be happy to do so.

(2) *Varying sources of empirical V1 data.*

R2.5 - *A real strength of the paper is robust examination of predicted model results in empirical monkey V1 data, but it is a bit confusing that the sources of data vary across sections. For example: surround stim reduces variability (natural images in 2 anesthetized monkeys), RF vs surround stim effects (natural images in 1 awake monkey; also gratings in 1 (or 2?) awake monkeys and gratings in 3 anesthetized monkeys), orientation selectivity (compound static gratings in 2 awake monkeys).*

It would be helpful if the authors clarified why they collected data in both anesthetized and awake preps, and addressed whether the monkey's level of awareness might be related to differences in data; specifically, it appears that size-dependent grating-driven suppression is larger for awake than anesthetized animals (Expt 3 vs Expt 2, table 1). Could this arise because, in the awake state, other factors can contribute to either uncertainty and/or mixer inference (e.g. attention)?

The reviewer is right that we used different preps for the different datasets. There were two motivations for this. First, we had a large set of previously collected anesthetized data, including data we had previously used to test the GSM (Coen-Cagli et al., 2015). We did not want readers to question why these previous data (highly relevant to the new questions addressed in this manuscript) were not used. Second, we collected data in both anesthetized and awake animals because the two preparations have complementary strengths. Awake fixating animals allow us to measure responses in cortex in its natural states, but there are small but non-negligible eye movements that can introduce stimulus-induced variability (because the retinal input changes within a fixation and across repetitions of the same image). In the anesthetized prep, this problem is mitigated because the animal is paralyzed and eye movements are minimized, therefore providing a control for the effects of eye movements. This is explained on page 8 lines 23-29. By observing that results in both preparations are consistent with the model predictions, we can be sure that the outcome was not determined by either brain state or by uncontrolled eye movements.

Regarding brain state, we agree with the reviewer that it likely affects response variability. We also agree that other signals (e.g. in attention) can influence variability, and these signals might differ between the anesthetized and awake state. However, it is not evident that this would explain the quantitative differences between our awake and anesthetized datasets, because it is not clear if fluctuations in signals unrelated to the stimulus (e.g. attention) are affected by stimulus manipulation (size) in the same way that stimulus-related uncertainty is. Conversely, if those other factors are not affected by stimulus size, then their net effect would be to partly mask the size-dependent reduction FF. This point relates to a limitation of our model, emphasized in Discussion page 11 40-45, i.e. that the GSM does not include latent variables to account for factors unrelated to the stimulus (e.g. cortical state or attention): all of the variability in our model is due to stimulus uncertainty, and changes in the stimulus thus lead to strong changes in variability.

(3) *Presentation of model FF effects.*

R2.6 - *There are a few places where model predictions concerning mean/variance relationships could be better conveyed. The authors state in the text their modeling results re: FF and mean response strength (pg 4, lines 31-33), but the model predictions are never actually quantified or shown as a figure. I think it would be helpful if the authors could add a figure panel - for example as Fig 1E - depicting model variance:mean characteristics as mean response changes. Also, in terms of the relationship between FF and surround stimulation, the authors state that in their empirical data "the vast majority (91.3%) had a lower FF for large images than small ones, consistent with model predictions" (pg 6 lines 24-25) - however, the paper doesn't show model predictions for FF and surround stimulation.*

Regarding the first point, we have expanded the simulation used for Fig. 1D, so that the dependence of FF on mean activity is more evident. In particular in our previous version we selected natural patches randomly, but many of these patches expressed the encoded feature (center vertical edge) weakly, if at all. Absent or ambiguous stimuli led to high variability at low rates. In our new version of Fig. 1D we first computed the phase-invariant signal strength on the center filter as $(x_{1+}^2 + x_{1-}^2)$, and then selected the 1,000 patches that were above the median of the full distribution (computed over 2,000 patches). This is equivalent to pre-selecting sufficiently responsive stimuli.

As suggested, we have also added a new Fig. 1E that shows the FF as a function of the mean response for those same stimuli.

The new figure caption reads:

DEL > (D) Mean versus variance of a GSM model neuron in response to 500 patches of natural images.

ADD p6.7-13> (D) Mean versus variance of a GSM model neuron in response to 1000 patches of natural images. Patches were selected randomly, with the requirement of sufficient signal strength inside the RF, i.e. above the median of the full distribution of $(x_1^2 + x_2^2)$ on natural scenes, where x_1 and x_2 are the odd and even phases of the center vertical filter (see Methods). (E) The Fano factor (ratio between mean and variance) increases with the mean response. The plot shows the FF as a function of mean for the same GSM simulation reported in (D). Red dashed line represents the best linear fit. Pearson corr. 0.214, $p < 10^{-4}$.

Regarding the second point, we have reported the model prediction in Fig. 2C and the text on page 6 line 38 “This result agrees qualitatively with the model (Fig. 2C, orange symbol).”

Other points

pg 4, line 2 - typo: “known as Poisson-like response statistics; known as Poisson-like response statistics”^[17]_[SEP]

We have now removed the term Poisson-like, as explained in response to R1.

pg 8, line 30 - “we also observed the same trends in data” I’d suggest using a different word than “trends”, since it has statistical connotations (specifically borderline significance) whereas the results are very significant

Thank you for this suggestion, we now have changed to:

DEL > Similar trends were evident across all recorded neurons for stimuli ranging from approximately half the RF size up to several times larger (N=86; Fig. 2F)

ADD p8.16-17 > Similar effects were evident across ...

DEL > We also observed the same trends [...] in data from anesthetized macaques.

ADD p8.26-27> New analyses of previously published data from anesthetized animals (Coen-cagli et al 2015) also confirmed these results (Table...).

pg 8 - It is unclear how many monkeys contribute to the awake grating data. The text says “we measured responses to static gratings both in the same animal and in a second awake monkey, and similar results” but the 19 neurons in Expt 2 are described in the Fig S4A legend as “one monkey”.

We have clarified as follows:

DEL > we measured responses to static gratings both in the same animal and in a second awake monkey, and obtained similar results...

ADD p8.25> we measured responses to static gratings in the same animal, and obtained similar results...

table 1 - The column headings with “FF change” are unclear since the term doesn’t specify a direction of change, would be better as “decrease”. Also, “vs.” doesn’t convey which is the comparison condition. The table legend clarifies it, but it would be easier if the headings themselves were clearer.

We have now clarified as follows:

DEL Table 1 > FF change X Vs. Y RF

ADD Table 1 > FF decrease (x RF) - (Y RF)

supp, line 56 - typo “ad FFs”

Thank you. Fixed.

Reviewers' Comments:

Reviewer #1:

Remarks to the Author:

I thank the authors for their responses. They have addressed all my points, and I only have two minor comments left.

1. The way the authors addressed the point that Reviewer 2 (rightly) raised about clarifying whether, and if so how, the model's predictions actually depend on natural image statistics got me thinking. Implicit (and now, thankfully, quite explicit) in the author's argument is that it is the structure of their GSM (rather than the precise setting of its parameters), namely the multiplicative interaction between the modulator and local features, that is 1. key to the results about the modulation of variability they obtain, 2. specific to natural image statistics, and 3. thus creates the much needed logical bridge between the two. Indeed, to drive point #1 home, they do the control experiment in which they show that a different model structure (with additive rather than multiplicative interaction between the modulator and local features) does not generate the same results about variability (and its modulations). However, I felt that they never quite did the same for #2. In other words, we "just" need to take it on belief that the GSM with the multiplicative interaction (mGSM) is a better model of natural image (patch) statistics than with an additive interaction (aGSM) — and that this is specific to natural image statistics and not to gratings or white noise. There may be previous results (eg from Simoncelli) showing this — but then this would need to be explicitly stated. If there aren't, then the authors should be able to show this. (I.e. obtain a double dissociation by doing some sort of model selection, showing that the mGSM is a better model of natural image statistics than the aGSM, but it is the other way around for grating and/or white noise.) I am fairly optimistic that this should be possible: the aGSM model is essentially identical to (a special case of) factor analysis, which is not sparse and has no power correlations between filters, and so should be a worse model of natural image statistics than the mGSM — while white noise is indeed not sparse and has no power correlations between filters and so should be better modeled by the aGSM than the mGSM.

2. This is probably just me being slow but I don't quite understand how Fig.1E can show that the "Fano factor (FF) increased for stimuli that elicited stronger mean responses" (p.4, lines 40-41) — whereas both basic manipulations that the authors now show in Fig. S9 (in response to a comment I made in the previous round), changing contrast and orientation, cause changes in responses such that higher mean responses (higher contrast, orientation closer to preferred) are accompanied by lower Fano factors. Unless this is not just a trivial oversight by me, it may be useful if the authors could comment on this where they describe these results.

Very minor points:

- p.4, line 13: "both the inferred value of g and its uncertainty" → perhaps it's worth making explicit that what you mean here is that the mean and the std of the posterior over g that you are talking about, so "both the inferred value of g and its uncertainty (i.e. the mean and s.t.d. of the posterior over g)"
- p.6, lines 17-18, and 28: "stimuli that provide more information about the modulator" and "by providing information about the global modulator" → To me, these sounded a bit misleading as if this was about the variance of the nu-posterior, whereas it's really about its mean. So I would suggest "stimuli that imply a higher estimate of the modulator" and "by resulting in a higher estimate of the global modulator", or something similar.
- p.12, line 34: "like changes in sparseness and reliability" → "such as changes in sparseness and reliability"

Mate Lengyel

Reviewer #2:

Remarks to the Author:

This is a revised paper from Festa et al examining the theoretical foundation for variability in neuron firing rates. They have been very responsive in their revisions (to both my comments and those of the other reviewer), and I am glad to endorse publication.

REVIEWER COMMENTS

Reviewer #1 (Remarks to the Author):

I thank the authors for their responses. They have addressed all my points, and I only have two minor comments left.

Thank you!

R1.1 The way the authors addressed the point that Reviewer 2 (rightly) raised about clarifying whether, and if so how, the model's predictions actually depend on natural image statistics got me thinking. Implicit (and now, thankfully, quite explicit) in the author's argument is that it is the structure of their GSM (rather than the precise setting of its parameters), namely the multiplicative interaction between the modulator and local features, that is 1. key to the results about the modulation of variability they obtain, 2. specific to natural image statistics, and 3. thus creates the much needed logical bridge between the two. Indeed, to drive point #1 home, they do the control experiment in which they show that a different model structure (with additive rather than multiplicative interaction between the modulator and local features) does not generate the same results about variability (and its modulations). However, I felt that they never quite did the same for #2. In other words, we "just" need to take it on belief that the GSM with the multiplicative interaction (mGSM) is a better model of natural image (patch) statistics than with an additive interaction (aGSM) — and that this is specific to natural image statistics and not to gratings or white noise. There may be previous results (eg from Simoncelli) showing this — but then this would need to be explicitly stated. If there aren't, then the authors should be able to show this. (I.e. obtain a double dissociation by doing some sort of model selection, showing that the mGSM is a better model of natural image statistics than the aGSM, but it is the other way around for grating and/or white noise.) I am fairly optimistic that this should be possible: the aGSM model is essentially identical to (a special case of) factor analysis, which is not sparse and has no power correlations between filters, and so should be a worse model of natural image statistics than the mGSM — while white noise is indeed not sparse and has no power correlations between filters and so should be better modeled by the aGSM than the mGSM.

Thanks for pointing this out. We have included a model comparison figure in Supplemental Figure S1 (this was Fig. S11 in the previous submission; new panels C,D), also copied below for completeness. In the figure, we assess the quality of an additive versus multiplicative GSM fit to natural images and to white noise (cross-validated log-likelihood). As expected, the multiplicative GSM captured natural image statistics better, due to the sparseness and power correlations. In case of white noise instead, the statistics are essentially Gaussian, and the presence of a multiplicative mixer hindered the model effectiveness at explaining the data.

We have added the following sentence in the main text, page 1 line 40: "First, we verified that the multiplicative effect of the modulator allows the GSM to capture the statistics of natural images (Wainwright, Simoncelli et al. 2000) better than an additive modulator (Supplementary Fig. S1)."

Model comparison: multiplicative Vs additive GSM

ADD Supplement , Figure S11 caption >

C,D. Model comparison between additive and multiplicative GSM for different image statistics. The plot reports the average log-likelihood over a set of 10,000 test image patches for a multiplicative and an additive GSM, both without additive noise. The measure is normalized so that 0 corresponds to the log-likelihood for a null model where each element of \mathbf{x} is an independent Normal. As expected, the multiplicative version of the GSM is better adapted to natural image statistics (Wainwright et al., 2000). White noise statistics, instead, can be captured by a multivariate Normal distributions, therefore the additive model, still Normal in nature, performs better than the multiplicative.

Methods. For these figures only, the generative model takes the form $\mathbf{x} = \nu .+ \mathbf{g} + \boldsymbol{\eta}$. The input filters are the same used for the GSM, however here the mixer ν is a scalar *additive* term, with $.+$ indicating the scalar-vector sum. Since a positive-only mixer might bias the posterior $P(\mathbf{x})$, we chose the prior $\nu \sim \mathcal{N}(0, \sigma_\nu)$ for the mixer. To train the model, the mixer was optimized to match the statistics of $\text{mean}(\mathbf{x})$ over natural image patches. The covariance structure of the features was trained using expectation maximization over 10,000 natural image patches, assuming a noiseless model. We then added a noise term with the same structure and scaling used for the GSM. Finally, to convert the latent feature variable into a spike count, we used: $r = \alpha(g_{1+} + \beta)$, with β sufficiently high to avoid negative terms. The phase of the input gratings was also regulated so that the filters output would be positive. For panels **C,D** we considered models without additive noise. We drew the training patches from 80% of the image dataset (which comprises 500 full scale images in total), and the test patches from the remaining 20%. Lastly, the models used in **D** were optimized using white noise patches instead of natural patches.

R1.2 This is probably just me being slow but I don't quite understand how Fig.1E can show that

the "Fano factor (FF) increased for stimuli that elicited stronger mean responses" (p.4, lines 40-41) — whereas both basic manipulations that the authors now show in Fig. S9 (in response to a comment I made in the previous round), changing contrast and orientation, cause changes in responses such that higher mean responses (higher contrast, orientation closer to preferred) are accompanied by lower Fano factors. Unless this is not just a trivial oversight by me, it may be useful if the authors could comment on this where they describe these results.

We appreciate this question, and agree that there is an apparent ambiguity. To shed light on this, we adopted a simplified 2D GSM model with no additive noise, where g_1 directly represents neural activation, while g_2 represents the surround influence. This model was already used in figure S6, to illustrate a simplified behavior for well-centered versus offset stimuli. To better illustrate the relation between mean and FF, here we considered three input levels -- very low contrast, medium contrast, strong contrast -- along with different configurations, based on whether the stimulus is only in the RF field (1st dimension), only in the surround region (2nd dimension), or in both. Stimuli are illustrated in the lower part of the figure.

When only contrast increased (a-b transitions in the plot below), the mean and the FF moved indeed in opposing directions, as also shown in S9. There are however many manipulations that cause rate and FF to change in the same direction. As a first example, for weak RF signal and strong surround, mean and FF are both low, due to the divisive influence of the surround (2a/2b). From this point, increasing the RF up to the same level as the surround (2 to 3 transition) caused an increase in FF, along with an increase in mean response (see also the role of x_1 in Eq. S25). As a second example, a reduction of surround stimulus starting from a full stimulus (3 to 4 transition) caused a decrease in surround normalization, which affected the variance more than the mean. Therefore, once again, mean and FF increased together.

Linking these modeling aspects to data would require further analyses and experiments, therefore we prefer not to include the figure in the publication. Nonetheless, we have added two clarifications in the text, as follows:

Page 5, line 4: "...Fano factor (FF), increased *on average* for stimuli that elicited stronger mean..."

Page 15 line 30: "Notice also that the expected value and the FF are not always monotonically related, because λ depends both on inputs inside and outside the RF, and appears with different exponents in the FF and expected value. For instance, surround stimulation affects only λ and thus changes the FF and expected value in the same direction, whereas changing contrast affects both numerator and denominator resulting in opposite scaling of the expected value and FF."

GSM model, 2D

Very minor points:

- p.4, line 13: "both the inferred value of g and its uncertainty" → perhaps it's worth making explicit that what you mean here is that the mean and the std of the posterior over g that you are talking about, so "both the inferred value of g and its uncertainty (i.e. the mean and s.t.d. of the posterior over g)"

Added on page 4 line 19

- p.6, lines 17-18, and 28: "stimuli that provide more information about the modulator" and "by providing information about the global modulator" → To me, these sounded a bit misleading as if this was about the variance of the nu-posterior, whereas it's really about its mean. So I would suggest "stimuli that imply a higher estimate of the modulator" and "by resulting in a higher estimate of the global modulator", or something similar.

DEL > Specifically, stimuli that provide more information about the modulator reduce uncertainty and thus should reduce response variability

ADD p7, line 12 > . Specifically, stimuli that lead to a higher estimate of the modulator present less uncertainty over the hidden feature, and thus should reduce response variability.

DEL > ... validating our intuition that surround stimuli reduce uncertainty by providing information about the global modulator.

ADD p7 , line 24 > ... validating our intuition that surround stimuli reduce uncertainty by resulting in a higher estimate of the global modulator

- p.12, line 34: "like changes in sparseness and reliability" → "such as changes in sparseness and reliability"

Added on p 13, line 43

Reviewer #2 (Remarks to the Author):

This is a revised paper from Festa et al examining the theoretical foundation for variability in neuron firing rates. They have been very responsive in their revisions (to both my comments and those of the other reviewer), and I am glad to endorse publication.

Thank you!

Reviewers' Comments:

Reviewer #1:

Remarks to the Author:

The authors have addressed all my comments. I thank them for their efforts and congratulate them for their paper.

Máté Lengyel